


# Large-eddy simulations with ClimateMachine v0.2.0: a new open-source code for atmospheric simulations on GPUs and CPUs

Akshay Sridhar[1], Yassine Tissaoui[2], Simone Marras[2], Zhaoyi Shen[1], Charles Kawczynski[1], Simon Byrne[1], Kiran Pamnany[1], Maciej Waruszewski[3], Thomas H. Gibson[4], Jeremy E. Kozdon[3], Valentin Churavy[5], Lucas C. Wilcox[3], Francis X. Giraldo[3], and Tapio Schneider[1,6]

[1]California Institute of Technology, Pasadena, California, USA
[2]New Jersey Institute of Technology, Newark, New Jersey, USA
[3]Naval Postgraduate School, Monterey, California, USA
[4]University of Illinois Urbana-Champaign, Urbana-Champaign, Illinois, USA
[5]Massachusetts Institute of Technology, Cambridge, Massachussetts, USA
[6]Jet Propulsion Laboratory, California Institute of Technology, Pasadena, California, USA

**Correspondence:** Akshay Sridhar (asridhar@caltech.edu)

**Abstract.** We introduce ClimateMachine, a new open-source atmosphere modeling framework using the Julia language to be performance portable on central processing units (CPUs) and graphics processing units (GPUs). ClimateMachine uses a common framework both for coarser-resolution global simulations and for high-resolution, limited-area large-eddy simulations (LES). Here, we demonstrate the LES configuration of the atmosphere model in canonical benchmark cases and atmospheric flows, using an energy-conserving nodal discontinuous-Galerkin (DG) discretization of the governing equations. Resolution dependence, conservation characteristics and scaling metrics are examined in comparison with existing LES codes. They demonstrate the utility of ClimateMachine as a modelling tool for limited-area LES flow configurations.

## 1 Introduction

Hybrid computer architectures and the need to exploit the power of graphics processing units (GPUs) are increasingly driving developments in atmosphere and climate modeling (e.g., Schalkwijk et al., 2012; Palmer, 2014; Schalkwijk et al., 2015; Marras et al., 2015; Abdi et al., 2017b, a; Fuhrer et al., 2018; Schär et al., 2020). The sheer computing power available on modern hardware architectures presents opportunities to accelerate atmosphere and climate modeling. However, exploiting this computing power requires re-coding atmosphere and climate models to an extent not seen in decades, and portable performance and scaling across different platforms remain difficult to achieve (Fuhrer et al., 2014; Balaji, 2021).

In this paper, we introduce ClimateMachine, a new open-source atmosphere model written in the Julia programming language (Bezanson et al., 2017) to provide a computational framework that is portable across CPU and GPU architectures. Additionally, the model is designed to be usable across a range of physical process scales, from large-eddy simulations (LES) with meter-scale resolution to global circulation models (GCM) with horizontal resolutions of tens of kilometers, as in a few other recent models (Dipankar et al., 2015). The use of Julia aims to increase accessibility and utility of ClimateMachine as a simulation tool. We focus on the LES configuration of ClimateMachine in this paper.





Since the pioneering work on LES by Smagorinsky (1963) and Lilly (1962), several models have been developed to improve the ability of LES to model atmospheric turbulence; from the extensive work by Deardorff in the 1970s and 1980s (Deardorff, 1970, 1974, 1976, 1980), by Moeng in the 1980s and beyond (Moeng, 1984; Moeng and Wyngaard, 1988; Sullivan et al., 1994; Moeng et al., 2003), to Stevens, Teixeira, Mellado, and others in the last two decades (Stevens et al., 2003, 2005; Savic-Jovcic

and Stevens, 2008; Matheou et al., 2011; Pressel et al., 2015; Matheou, 2016; Matheou and Teixeira, 2019; Mellado, 2017; Mellado et al., 2018). LES results in canonical flows are sensitive to the fine details of the equations used to represent the flow dynamics, the viscous dissipation, the thermodynamics, and the numerical methods used to solve them (Ghosal, 1996; Chow and Moin, 2003; Kurowski et al., 2014), especially in the case of cloud simulations (Stevens et al., 2005; Siebesma et al., 2003; Schalkwijk et al., 2012, 2015; Schneider et al., 2019; Pressel et al., 2015, 2017).

One distinguishing aspect of the ClimateMachine LES is that it uses a nodal discontinuous Galerkin (DG) formulation to approximate the Navier-Stokes equations for compressible flow (Giraldo et al., 2002; Hesthaven and Warburton, 2008a; Giraldo and Restelli, 2008; Kopriva, 2009; Kelly and Giraldo, 2012; Giraldo, 2020). The DG method is a spectral-element generalization of finite-volume methods. It lends itself well to modern high-performance computing architectures because its communication overhead is low, enabling scaling on manycore processors including GPUs (Abdi et al., 2017b). Another im-

portant consideration within ClimateMachine is the use of total energy of moist air as a prognostic variable, ensuring energetic consistency of the simulations. We demonstrate that the ClimateMachine LES can be successfully used to simulate canonical LES benchmarks, including simulations of flows over mountains and different cloud and boundary-layer regimes (e.g., Straka et al., 1993; Schär et al., 2002; Stevens et al., 2005).

In what follows, we describe the conceptual and numerical foundations and governing equations of ClimateMachine and

demonstrate the model in a set of standard two- and three-dimensional benchmark simulations. Section 2 begins by highlighting the governing equations. Their numerical approximation through the DG representation is described in Section 3. Section 4 presents sub-grid scale models used in the LES to represent under-resolved flow physics, with results from key benchmarks presented in Section 5. Conservation properties are examined in Section 6, and performance on CPU and GPU hardware is described in Sections 7 and 8, respectively. Section 9 contains closing remarks. Additional details about the model, boundary

conditions, statistical definitions, and computer hardware are summarized in the appendices.

## 2 Governing Equations

### 2.1 Working fluid

The working fluid of the atmosphere model is moist, potentially cloudy air, considered to be an ideal mixture of dry air, water vapor, and condensed water (liquid and ice) in clouds. Dry air and water vapor are taken to be ideal gases. The specific volume

of the cloud condensate is neglected relative to that of the gas phases (it is a factor $10^3$ less than that of the gas phases). All gas phases are assumed to have the same temperature, and are advected with the same velocity $\boldsymbol{u} = (u, v, w)^T$. Cloud condensate is assumed to sediment relative to gaseous phases slowly enough to be in thermal equilibrium with the surrounding fluid.





The density of the moist air is denoted by $\rho$. We use the following notation for the mass fractions of the moist air mixture (mass of a constituent divided by the total mass of the working fluid):

  – $q_d$: dry air mass fraction,

  – $q_v$: water vapor specific humidity,

  – $q_l$: liquid water specific humidity,

  – $q_i$: ice specific humidity,

  – $q_c = q_l + q_i$: condensate specific humidity,

  – $q_t = q_v + q_c$: total specific humidity.

Because this enumerates all constituents of the working fluid, we have $q_t + q_d = 1$. In Earth's atmosphere, the water vapor specific humidity $q_v$ dominates the total specific humidity $q_t$ and is usually $\mathcal{O}(10^{-2})$ or smaller; the condensate specific humidity is typically $\mathcal{O}(10^{-4})$. Hence, water is a trace constituent of the atmosphere, and only a small fraction of atmospheric water is in condensed phases. The working fluid pressure is the sum of the partial pressures of dry air and water vapor such
that $p = \rho(R_d q_d + R_v q_v)T$, where $R_d$ is the specific gas constant of dry air, and $R_v$ is the specific gas constant of water vapor.

### 2.2 Mass balance

Moist air mass satisfies the conservation equation

$$\frac{\partial \rho}{\partial t} + \nabla \cdot (\rho \boldsymbol{u}) = \rho \hat{\mathcal{S}}_{q_t}. \tag{1}$$

Moist air mass is not exactly conserved where precipitation forms, sublimates, or evaporates, where water diffuses, or where
condensate sediments relative to the gas phases (Bott, 2008; Romps, 2008). The right-hand side involves the local source/sink of water mass $\hat{\mathcal{S}}_{q_t}$ owing to such non-conservative processes, which we take into account although it is small because water is a trace constituent of the atmosphere.

### 2.3 Total water balance

Total water satisfies the balance equation

$$\frac{\partial(\rho q_t)}{\partial t} + \nabla \cdot (\rho q_t \boldsymbol{u}) = \rho \mathcal{S}_{q_t} - \nabla \cdot (\rho \boldsymbol{d}_{q_t}) + \nabla \cdot (\rho q_c w_c \hat{\boldsymbol{k}}) \equiv \rho \hat{\mathcal{S}}_{q_t}. \tag{2}$$

Here, the source/sink $\mathcal{S}_{q_t}$ arises from evaporation or sublimation of precipitation and formation of precipitation. Diffusive fluxes of moisture are captured by $\boldsymbol{d}_{q_t}$. The effective sedimentation velocity of cloud condensate $w_c$ is defined such that

$$q_c w_c = q_l w_l + q_i w_i \tag{3}$$

with $w_l$ and $w_i$ defined to be positive downward ($\hat{\boldsymbol{k}}$ being the upward pointing unit vector). The right-hand side $\rho \hat{\mathcal{S}}_{q_t}$ of the
total water balance equation is the same as the right-hand side of the mass balance equation (1).





## 2.4 Momentum balance

The coordinate independent form of the conservation law for momentum is

$$
\frac{\partial(\rho \boldsymbol{u})}{\partial t} + \nabla \cdot [\rho \boldsymbol{u} \otimes \boldsymbol{u} + (p - p_r)\boldsymbol{I}_3] = -(\rho - \rho_r)\boldsymbol{\nabla}\Phi - 2\boldsymbol{\Omega} \times \rho\boldsymbol{u}
$$

$$
- \nabla \cdot (\rho\boldsymbol{\tau}) - \nabla \cdot (\boldsymbol{d}_{q_t} \otimes \rho\boldsymbol{u}) + \nabla \cdot \left(q_c w_c \hat{\boldsymbol{k}} \otimes \rho\boldsymbol{u}\right) + \rho\boldsymbol{F_u}, \quad (4)
$$

where $\boldsymbol{I}_3$ is the rank-3 identity matrix, $\Phi$ is the effective gravitational potential including centrifugal accelerations, $\boldsymbol{\tau}$ is a viscous and/or subgrid-scale (SGS) momentum flux tensor; and $\boldsymbol{F_u}$ (typically with $\boldsymbol{F_u} \cdot \boldsymbol{u} < 0$, so that $\boldsymbol{F_u}$ represents a momentum sink) is any other drag force per unit mass that may be applied, for example, at the lower boundary. The term involving the planetary angular velocity $\boldsymbol{\Omega}$ accounts for Coriolis forces. To improve numerical stability, we have factored out a reference

state with a pressure $p_r(z)$ and density $\rho_r(z)$ that depend only on altitude $z$ and are in hydrostatic balance, so that they satisfy

$$
\boldsymbol{\nabla}p_r = -\rho_r\boldsymbol{\nabla}\Phi.
$$

The tensor involving the diffusive flux $\boldsymbol{d}_{q_t}$ of water on the right-hand side of (4) represents the momentum flux carried by water that is diffusing; this term is usually very small, but we take it into account.

## 2.5 Energy balance

The specification of a thermodynamic or energy conservation equation closes the equations of motion for the working fluid. We use the total specific energy, $e^{\text{tot}}$, as the prognostic variable. Total energy is conserved in reversible moist processes such as phase transitions of water.

Total energy satisfies the conservation law (Romps, 2008; Bott, 2008)

$$
\frac{\partial(\rho e^{\text{tot}})}{\partial t} + \nabla \cdot ((\rho e^{\text{tot}} + p)\boldsymbol{u}) = -\nabla \cdot (\rho\boldsymbol{F}_R) - \nabla \cdot [\rho(\boldsymbol{J} + \boldsymbol{D})] + \rho Q + \nabla \cdot \left(\rho W_c \hat{\boldsymbol{k}}\right) - \nabla \cdot (\boldsymbol{u} \cdot \rho\boldsymbol{\tau}) - \sum_{j \in \{v, l, i\}} (I_j + \Phi)\rho C(q_j \to q_p) - M,
$$

(5)

where the total specific energy $e^{\text{tot}}$ is defined by

$$
e^{\text{tot}} = \frac{1}{2}\|\boldsymbol{u}\|^2 + \Phi + I. \tag{6}
$$

The constituents (dry air and moisture components) here are assumed to be moving with the same velocity $\boldsymbol{u}$ (that is, we neglect, as is common, the diffusive and sedimentation fluxes of water in the kinetic energy). The constituents are also assumed

to be in thermal equilibrium at the same temperature $T$, so that the specific internal energy of moist air is the weighted sum of the specific energies of the constituents: dry air ($I_d$), water vapor ($I_v$), liquid water ($I_l$), and ice ($I_i$):

$$
I(T, q) = (1 - q_t)I_d(T) + q_v I_v(T) + q_l I_l(T) + q_i I_i(T), \tag{7}
$$





**Table 1.** Thermodynamic constants in CLIMAParameters

| | |
|---|---:|
| $R_d$ | 287 J $(\text{kg K})^{-1}$ |
| $R_v$ | 462 J $(\text{kg K})^{-1}$ |
| $c_{vd}$ | 717.5 J $(\text{kg K})^{-1}$ |
| $c_{vv}$ | 1397.5 J $(\text{kg K})^{-1}$ |
| $c_{vl}$ | 4181 J $(\text{kg K})^{-1}$ |
| $c_{vi}$ | 2100 J $(\text{kg K})^{-1}$ |
| $T_0$ | 273.16 K |
| $L_{v,0}$ | $2.5008 \times 10^6$ J $\text{kg}^{-1}$ |
| $L_{f,0}$ | $0.3336 \times 10^6$ J $\text{kg}^{-1}$ |

with

$$I_d(T) = c_{vd}(T - T_0), \tag{8a}$$

$$I_v(T) = c_{vv}(T - T_0) + I_{v,0}, \tag{8b}$$

$$I_l(T) = c_{vl}(T - T_0), \tag{8c}$$

$$I_i(T) = c_{vi}(T - T_0) - I_{i,0}. \tag{8d}$$

Here, $c_{vk}$ for $k \in \{d, v, l, i\}$ are isochoric specific heat capacities for the appropriate species denoted by $k$; they are taken to be constant. The reference specific internal energy $I_{v,0}$ is the difference in specific internal energy between vapor and liquid at the arbitrary reference temperature $T_0$; $I_{i,0}$ is the difference in specific internal energy between ice and liquid at $T_0$ (Romps, 2008). The reference internal energies are related to specific latent heats of vaporization and fusion, $L_{v,0}$ and $L_{f,0}$, at the reference temperature $T_0$ through

$$I_{v,0} = L_{v,0} - R_v T_0, \tag{9}$$

$$I_{i,0} = L_{f,0}. \tag{10}$$

The values of the thermodynamic constants we use are listed in Table 1.

Furthermore, the flux $\boldsymbol{F}_R$ is the radiative energy flux per unit mass; $\boldsymbol{J}$ is the conductive energy flux per unit mass, and $\boldsymbol{D}$ is the specific enthalpy flux associated with the diffusive flux of water

$$\boldsymbol{D} = (e_v^{\text{tot}} + R_v T)\boldsymbol{d}_{q_v} + e_l^{\text{tot}}\boldsymbol{d}_{q_l} + e_i^{\text{tot}}\boldsymbol{d}_{q_i}. \tag{11}$$

The flux $\boldsymbol{u} \cdot \rho\boldsymbol{\tau}$ is the energy flux associated with the viscous and/or SGS turbulent momentum flux; and $Q$ is any internal energy source (e.g., external diabatic heating). The flux

$$W_c = q_l e_l^{\text{tot}} w_l + q_i e_i^{\text{tot}} w_i \tag{12}$$





represents the downward energy flux due to sedimenting condensate.

The terms involving $\rho C(q_j \to q_p)$ $(j \in \{v, l, i\})$ represent the loss of internal and potential energy of moist air masses owing to precipitation formation; the kinetic energy loss is neglected, consistent with the neglect of the source/sink associated with precipitation formation in the momentum balance (4). Additional energy sinks involve the energy loss owing to heat transfer from the working fluid to precipitation as it falls through air and possibly melts at the freezing level (Raymond, 2013); the associated energy sources/sinks are generally provided by a microphysics parameterization and are subsumed in the term $M$.

## 2.6 Equation of state

Pressure $p$ is calculated from the ideal-gas law

$$p = \rho R_m T, \tag{13}$$

where $R_m$ is the gas "constant" of moist air,

$$R_m(q) = R_d(1 - q_t) + R_v q_v$$
$$= R_d \left[ 1 + (\varepsilon_{dv} - 1) q_t - \varepsilon_{dv} q_c \right], \tag{14}$$

with the ratio of the gas constants of water vapor and of dry air $\varepsilon_{dv} = R_v / R_d$.

## 2.7 Saturation adjustment

Gibbs' phase rule states that in thermodynamic equilibrium, the temperature $T$ and liquid and ice specific humidities $q_l$ and $q_i$ can be obtained from the three thermodynamic state variables density $\rho$, total water specific humidity $q_t$, and internal energy $I$. Thus, the above equations suffice to completely specify the thermodynamic state of the working fluid, given $\rho$, $q_t$, and $I$, the latter obtained from the total energy via its definition (6).

Obtaining the temperature and condensate specific humidities from the state variables $\rho$, $q_t$, and $I$ is the problem of finding the root $T$ of

$$I^*(T; \rho, q_t) - I = 0, \tag{15}$$

where $I^*(T; \rho, q_t)$ is the internal energy at equilibrium, when the air is either unsaturated and there is no condensate ($q_v = q_t$), or water vapor is in saturation and the saturation excess $q_t - q_v$ is apportioned, according to temperature $T$, among the condensed phases $q_l$ and $q_i$. We solve this nonlinear "saturation adjustment" problem by Newton iterations with analytical gradients (cf. Tao et al., 1989; Pressel et al., 2015). To obtain the saturation vapor pressure and derived functions needed in this calculation, we assume all isochoric heat capacities to be constant (i.e., we assume the gases to be calorically perfect); with this assumption, the Clausius-Clapeyron can be integrated analytically, resulting in a closed-form expression for the saturation vapor pressure (Romps, 2008).

This procedure allows the use of total moisture $q_t$ as the sole prognostic variable, but confines the system to the assumption of equilibrium thermodynamics. Alternatively, using explicit tracers for the condensate specific humidities $q_l$ and $q_i$ allows non-equilibrium thermodynamics to be considered and mixed-phase processes to be explicitly modeled.





## 3 Discretization of the governing equations

### 3.1 Space discretization

The governing equations are discretized in space via a nodal DG approximation. To describe the DG procedure, we recast the equations (1)–(5) in compact notation as

$$\frac{\partial \boldsymbol{Y}}{\partial t} = -\nabla \cdot (\boldsymbol{F}_1 + \boldsymbol{F}_2) + \boldsymbol{\mathcal{S}}(\boldsymbol{Y}), \tag{16}$$

where $\boldsymbol{Y} = [\rho, \rho\boldsymbol{u}, \rho e^{\mathrm{tot}}, \rho q_t]^T$ is an abstract vector of state variables; $\boldsymbol{F}_1$ contains the fluxes not involving gradients of state variables and functions thereof; $\boldsymbol{F}_2$ contains the fluxes involving gradients of state variables (e.g., diffusive fluxes); and $\boldsymbol{\mathcal{S}}(\boldsymbol{Y})$ contains the sources.

The DG solution of (16) is approximated on the finite-dimensional counterpart $\Omega^h$ of the flow domain $\Omega$, which consists of $N_{\Omega_e}$ non-overlapping hexahedral elements $\Omega_e$ such that

$$\Omega^h = \bigcup_{e=1}^{N_{\Omega_e}} \Omega_e,$$

where a superscript $h$ indicates the discrete analog of a continuous quantity. By virtue of tensor-product operations allowed on hexahedral elements and the ability to rely on inexact quadrature when elements of order greater than 3 are utilized, high-order Galerkin methods are particularly attractive for operation intensive solutions (Kelly and Giraldo, 2012). Within each element, the finite dimensional approximation of $\boldsymbol{Y}(\mathbf{x}, t)$ is given by the expansion

$$\boldsymbol{Y}_e^h(\mathbf{x}, t) = \sum_{\alpha=1}^{(N+1)^3} \psi_\alpha^e(\mathbf{x}) \boldsymbol{Y}_\alpha^e(t), \tag{17}$$

where $(N+1)^3$ is the number of collocation points within the three-dimensional element of order $N$, and $\psi_\alpha^e$ are the interpolation polynomials evaluated at local point $\alpha$ inside element $e$.

From now on, the subscript/superscript $e$ is omitted with the understanding that all operations are executed element-wise unless otherwise stated. Furthermore, the physical elements in the $\mathbf{x} = (x, y, z)$ space are mapped to a reference element $\boldsymbol{\xi} = (\xi, \eta, \zeta)$. The three-dimensional basis functions $\psi_\alpha$ result from the one-dimensional functions $L_i(\xi)$, $L_j(\eta)$, and $L_k(\zeta)$ as the tensor product:

$$\psi_\alpha(\boldsymbol{\xi}) = L_i(\xi) \otimes L_j(\eta) \otimes L_k(\zeta), \qquad \forall i, j, k = 1, ..., N+1.$$

Each function $L$ is a one-dimensional (1D) Lagrange polynomial defined on the 1D reference element $[-1, 1]$. The Lagrange function evaluated at points $i$ along the $\xi$ direction within the element is

$$L_i(\xi) = \prod_{l=1, l \neq i}^{N+1} \frac{\xi - \xi_l}{\xi_i - \xi_l},$$





where $\xi_i$ are the $N+1$ co-located interpolation points along $\xi$. The polynomials $L_j$ and $L_k$ in the two other directions $\eta$ and $\zeta$ are built in the same way. The $N+1$ interpolation points may be chosen in variety of ways (Deville et al., 2002; Karniadakis and Sherwin, 1999); here we choose Legendre-Gauss-Lobatto (LGL) points (Giraldo and Restelli, 2008). The Kronecker $\delta$ property of the Lagrange polynomials is such that

$$\psi_i(\xi_l) = \delta_{il}$$

in 1D which, in three-dimensions (3D), translates to

$$\Psi_\alpha(\xi_a, \eta_b, \zeta_c) = \delta_{ia} \otimes \delta_{jb} \otimes \delta_{kc}. \tag{18}$$

This allows us to reduce the operation count, as follows.

We construct the space and time derivatives as

$$\frac{\partial \boldsymbol{Y}^h(\mathbf{x},t)}{\partial \mathbf{x}} = \sum_{\alpha=1}^{(N+1)^3} \frac{\partial \psi_\alpha(\mathbf{x})}{\partial \mathbf{x}} \boldsymbol{Y}_\alpha(t), \tag{19}$$

$$\frac{\partial \boldsymbol{Y}^h(\mathbf{x},t)}{\partial t} = \sum_{\alpha=1}^{(N+1)^3} \psi_\alpha(\mathbf{x}) \frac{\partial \boldsymbol{Y}_\alpha(t)}{\partial t}. \tag{20}$$

By virtue of the 3D Kronecker $\delta$ property, the spatial derivatives of the basis functions appearing here are given by

$$\frac{\partial \psi_\alpha}{\partial \xi}(\xi_a, \eta_b, \zeta_c) = \frac{\partial \psi_i(\xi_a)}{\partial \xi} \otimes \delta_{jb} \otimes \delta_{ck}, \tag{21}$$

$$\frac{\partial \psi_\alpha}{\partial \eta}(\xi_a, \eta_b, \zeta_c) = \delta_{ia} \otimes \frac{\partial \psi_j(\eta_b)}{\partial \eta} \otimes \delta_{ck}, \tag{22}$$

$$\frac{\partial \psi_\alpha}{\partial \zeta}(\xi_a, \eta_b, \zeta_c) = \delta_{ia} \otimes \delta_{jb} \otimes \frac{\partial \psi_k(\zeta_c)}{\partial \zeta}. \tag{23}$$

Using this property reduces the operation count significantly since we only require $3N$ operations instead of the $N^3$ operations otherwise needed to compute the derivatives at a given node (Abdi et al., 2017b).

The operators defined on the reference elements are mapped onto the physical space by means of the transformation

$$\nabla \psi = \mathbf{J}^{-1} \widehat{\nabla} \psi \tag{24}$$

where $\nabla = (\partial_x, \partial_y, \partial_z)^T$, $\widehat{\nabla} = (\partial_\xi, \partial_\eta, \partial_\zeta)^T$ and

$$\mathbf{J}^{-1} = \begin{bmatrix} \xi_x & \xi_y & \xi_z \\ \eta_x & \eta_y & \eta_z \\ \zeta_x & \zeta_y & \zeta_z \end{bmatrix}$$

is the inverse Jacobian of the transformation from physical space to the reference element.





The DG approximation of the differential equations (16) is constructed by multiplying, within each element, the equation by the test function $\psi_\alpha$ and then integrating over the element volume $\Omega_e$, such that

$$\int\limits_{\Omega_e} \psi_\alpha \left(\partial_t \boldsymbol{Y} + \nabla \cdot \boldsymbol{F}_1(\boldsymbol{Y}) + \nabla \cdot \boldsymbol{F}_2(\boldsymbol{Y}, \nabla \boldsymbol{Y})\right) d\Omega_e$$

$$= \int\limits_{\Omega_e} \psi_\alpha \boldsymbol{S}(\boldsymbol{Y}) d\Omega_e, \quad (25)$$

where $\psi_\alpha$ within each element belongs to the function space of square integrable piecewise polynomials of order $N$ (i.e., $\psi \in L^2$). By definition, these functions are discontinuous across element boundaries; differentiability is not globally required but only within each element (Hesthaven and Warburton, 2008b). Integrating the divergence term by parts yields

$$\int\limits_{\Omega_e} \Psi_\alpha \partial_t \boldsymbol{Y} \, d\Omega_e + \oint\limits_{\Gamma_e} \Psi_\alpha \mathbf{n} \cdot \boldsymbol{F}_1^*(\boldsymbol{Y}) \, d\Gamma_e$$

$$- \int\limits_{\Omega_e} \nabla \Psi_\alpha \cdot \boldsymbol{F}_1(\boldsymbol{Y}) \, d\Omega_e - \int\limits_{\Omega_e} \Psi_\alpha \nabla \cdot \boldsymbol{F}_2(\boldsymbol{Y}, \boldsymbol{\nabla Y}) \, d\Omega_e$$

$$= \int\limits_{\Omega_e} \Psi_\alpha \boldsymbol{S}(\boldsymbol{Y}) \, d\Omega_e, \quad (26)$$

where $\Omega_e$ and $\Gamma_e$ are, respectively, the volume and boundary of each element, $\mathbf{n}$ is the outward facing unit vector orthogonal to each element face, and $\boldsymbol{F}_1^*$ is a numerical flux. The imposition of the numerical fluxes across element boundaries is the numerical mechanism that promotes continuity of the discontinuous solution across the elements. The numerical fluxes are

calculated as the approximate solution to a Riemann problem across two neighboring elements. ClimateMachine currently implements the Rusanov (1961), Roe (1981), and Harten-Lax-van Leer-Contact (HLLC) (E. F. Toro et al., 1994; Harten, 1983) numerical fluxes. The Rusanov flux, for instance, is constructed as

$$\mathbf{n} \cdot \boldsymbol{F}_1^*(\boldsymbol{Y}) = \frac{\mathbf{n}}{2} \cdot \left[\boldsymbol{F}_1(\boldsymbol{Y}^-) + \boldsymbol{F}_1(\boldsymbol{Y}^+)\right] + \mathbf{n}\lambda_{\Gamma_e}\left(\boldsymbol{Y}^- - \boldsymbol{Y}^+\right), \quad (27)$$

where $\boldsymbol{Y}^-$ is the state at the internal interface of element $e$, $\boldsymbol{Y}^+$ is the state at the external interface of $e$, and $\lambda_{\Gamma_e}\left(\boldsymbol{Y}^-, \boldsymbol{Y}^+\right)$ is

an estimate of the maximum flow speed (e.g., the maximum eigenvalue of the Jacobian of the flux $\boldsymbol{F}_1$ with respect to the state variables, which is the speed of sound).

Because the second-order derivatives in $\nabla \cdot \boldsymbol{F}_2$ cannot be directly built with the weak variational formulation if a discontinuous function space is used (Bassi and Rebay, 1997), an auxiliary variable $\widetilde{\mathbf{Y}}$ is introduced such that

$$\nabla \mathbf{Y} = \widetilde{\mathbf{Y}} \quad (28)$$

$$\nabla \cdot (\mu \nabla \mathbf{Y}) = \nabla \cdot (\mu \widetilde{\mathbf{Y}}), \quad (29)$$

which can then be discretized via DG as

$$\int\limits_{\Omega_e} \Psi_\alpha \nabla \cdot \nabla \boldsymbol{Y} \, d\Omega_e \approx \oint\limits_{\Gamma_e} \Psi_\alpha \mathbf{n} \cdot \left(\mu \widetilde{\mathbf{Y}}^* - \mu \widetilde{\mathbf{Y}}\right) \, d\Gamma_e + \int\limits_{\Omega_e} \Psi_\alpha \nabla \cdot \left(\mu \widetilde{\mathbf{Y}}\right) \, d\Omega_e. \quad (30)$$





Here, $\widetilde{\mathbf{Y}}^*$ is approximated via centered flux like in Bassi and Rebay (1997). We also refer to Abdi et al. (2017b) for more details.

For algorithmic efficiency, inexact quadrature is used to calculate the integrals above. By virtue of inexact integration and of equations (17), (19), and (24), the variational DG equations yield the semi-discrete matrix problem

$$\frac{\mathrm{d}\boldsymbol{Y}_\mathrm{i}^\mathrm{e}}{\mathrm{d}t} = - \left(\nabla_{j,i}^T\right)\boldsymbol{F}_j^e + \boldsymbol{\mathcal{S}}_i^e + \frac{w_i{}^s|\mathbf{J}|_i^s}{w_i{}^e|\mathbf{J}|_i^e}\boldsymbol{n}_i^s\left(\boldsymbol{F}^e - \boldsymbol{F}^*\right)_i, \tag{31}$$

where $w_i^s$ are interpolation weights. The algebraic details to obtain this expression can be found in Giraldo and Restelli (2008), where the $s$ superscript indicates a value that is defined on the element boundary surface. The system (31) is integrated on each
element with respect to time.

In order to achieve good parallel scaling it is necessary to overlap communication and computation to the fullest extent possible. With DG (and all element-based Galerkin methods) this can be naturally achieved by splitting Eq. (31) into terms that arise from the approximation of volume integrals and surface integrals. All volume contributions can be calculated independently of element-to-element communication regardless of the order of the spatial approximation, as can surface integrals that
are not on elements which share boundaries across MPI ranks. Thus, in the code, we start with message passing interface (MPI) communication, do all volume calculations and surface calculations for elements not on boundaries shared across ranks, and then apply surface calculations for elements on the rank boundaries after communication operations have been completed. This approach makes DG naturally effective with respect to parallel computing as previously shown by, e.g., Müller et al. (2018) on CPUs, Abdi et al. (2017b) on GPUs. At high order, element-based Galerkin methods such as DG require fewer neighboring
degrees of freedoms than high-order finite difference and finite volume discretizations.

### 3.2   Time discretization

ClimateMachine provides a suite of time-integrators consisting of explicit Runge-Kunge methods, low-storage (Carpenter and Kennedy, 1994; Niegemann et al., 2012), strong stability-preserving (Shu and Osher, 1988) and additive Runge-Kutta (ARK) implicit-explicit (IMEX) methods (Giraldo et al., 2013; Kennedy and Carpenter, 2019).
The benchmarks presented in this paper with isotropic grid spacing are run using the 4th-order 14-stage method of Niegemann et al. (2012), which has a large explicit time-stepping stability region. One of the benchmarks, however, uses a highly anisotropic grid, which benefits from the use of a 1-D IMEX approximation; there we use a variant of the horizontally explicit, vertically implicit (HEVI) schemes by Bao et al. (2015). While 3-D IMEX is also an option, its performance in terms of time-to-solution is ultimately limited by the availability of scalable 3-D implicit solver algorithms.

### 4   Sub-grid scale models

The governing equations are resolved with the discretizations presented in Section 3. This leaves unresolved but dynamically significant scales on the computational grid that must be modeled with three-dimensional SGS models. In general, SGS fluxes are modeled as diffusive fluxes, which capture down-gradient transport of conservable scalar quantities assuming that mixing





lengths are small compared with the scales over which the gradients of the scalars vary. We address the physical form of
the diffusive flux components in equations (1)–(5), following which we describe standard models of subgrid-scale turbulence
available for use in ClimateMachine.

The diffusive momentum flux tensor $\boldsymbol{\tau}$ is represented in terms of the symmetric rate of strain tensor $\boldsymbol{S}$ such that

$$\boldsymbol{S}(\boldsymbol{u}) = \frac{1}{2}\left(\boldsymbol{\nabla}\boldsymbol{u} + (\boldsymbol{\nabla}\boldsymbol{u})^T\right),\tag{32}$$

with

$$\boldsymbol{\tau} = -(2\boldsymbol{\nu}_t\boldsymbol{S}).\tag{33}$$

Here, $\boldsymbol{\nu}_t$ is a turbulent viscosity tensor whose components are typically orders of magnitude larger than the molecular viscosity
and are a function of the velocity gradient tensor.

The diffusive flux $\boldsymbol{d}_{q_t}$ of total water specific humidity in equation 2 is modeled as

$$\boldsymbol{d}_{q_t} = -(\boldsymbol{\mathcal{D}_t}\boldsymbol{\nabla}q_t),\tag{34}$$

where $\boldsymbol{\mathcal{D}}_t$ is a turbulent diffusivity vector. The turbulent diffusivity $\boldsymbol{\mathcal{D}}_t$ is related to the turbulent viscosity tensor $\boldsymbol{\nu}_t$ via the
turbulent Prandtl number such that

$$\boldsymbol{\mathcal{D}}_t = \frac{\operatorname{diag}(\boldsymbol{\nu}_t)}{\operatorname{Pr}_t},\tag{35}$$

where $\operatorname{Pr}_t$ takes a typical value of $\operatorname{Pr}_t = 1/3$.

The unresolved flux of total enthalpy $h^{\mathrm{tot}}$ results in a diffusive subgrid flux term of the form

$$\boldsymbol{J} + \boldsymbol{D} = -(\boldsymbol{\mathcal{D}_t}\boldsymbol{\nabla}h^{\mathrm{tot}}),\tag{36}$$

where $\boldsymbol{J}$ is the thermal diffusion flux analogous to the molecular conductive heat flux, and $\boldsymbol{D}$ is the energy flux carried by
water vapor, defined in equation 11. For energetic consistency, we use the same turbulent diffusivity $\boldsymbol{\mathcal{D}_t}$ for moist enthalpy and
water.

## 4.1 Smagorinsky-Lilly model

The turbulent eddy viscosity $\nu_t$ in the model by Smagorinsky (1963) and Lilly (1962) (SL henceforth) is defined by means of
the magnitude of the rate of the strain tensor $\boldsymbol{S}$, whose components are $S_{ij} = \frac{1}{2}\left(\frac{\partial u_i}{\partial x_j} + \frac{\partial u_j}{\partial x_i}\right)$, according to

$$\nu_t = (C_s\Delta)^2\sqrt{2S_{ij}S_{ij}},\tag{37}$$

for $i,j = 1,2,3$; $C_s$ is a constant Smagorinsky coefficient usually within the range $0.12 < C_s < 0.21$; and $\Delta$ is the LES filter-
width. Inside each hexahedral element of order $N$ and side lengths $L_{x,y,z}$ along the $x,y,z$ directions, the effective grid resolu-
tion is $\Delta(x,y,z) = L_{x,y,z}/N$, which is the average distance between two consecutive nodal points. We use an isotropic eddy
viscosity tensor $\boldsymbol{\nu_t}$ in LES, with its components defined by equation (37).





## 4.2 Vreman eddy viscosity model

The SGS model developed by Vreman (2004) is of interest because of its robustness across flow regimes and because it has low dissipation near wall boundaries and in transitional flows. Its computational complexity is similar to the classical SL model.
While the Vreman model is extensively used in engineering LES, it is uncommon in atmospheric flows, where a constant coefficient SL or the 1-equation TKE model by Deardorff (1970, 1980) are the most common choices (see, e.g., Stevens et al. (2005)).

The turbulent eddy viscosity of this model depends on first-order derivatives of velocities and is given by

$$\nu_t = 2.5 C_s^2 \sqrt{\frac{B_\beta}{u_{i,j} u_{i,j}}}, \tag{38a}$$

where

$$B_\beta = \beta_{11}\beta_{22} + \beta_{11}\beta_{33} + \beta_{22}\beta_{33} - (\beta_{13}^2 + \beta_{12}^2 + \beta_{23}^2), \tag{38b}$$

$$\beta_{ij} = \Delta_m^2 u_{i,m} u_{j,m}, \tag{38c}$$

$$u_{i,j} = \frac{\partial u_i}{\partial x_j}. \tag{38d}$$

Here, summation over repeated indices $i, j \in \{1, 2, 3\}$ is implied, $C_s$ is the constant Smagorinsky coefficient, and $u_i$ represent
the components of the resolved-scale velocity vector, so that $u_{i,j}$ is the velocity gradient tensor. The mixing lengths $\Delta_m$ can be determined as the grid spacing in the direction implied by subscript $m$; in this paper $\Delta_m \equiv L_{x,y,z}/N$.

**Richardson correction in stable regions of the atmosphere** To account for the atmospheric stability and, effectively, reduce turbulence generation to zero in stably stratified atmospheres, we multiply the eddy viscosity by the correction factor

$$f_b = \begin{cases} 1 & \mathrm{Ri} \leq 0, \\ \max(0, 1 - \mathrm{Ri}/\mathrm{Pr}_t)^{1/4} & \mathrm{Ri} > 0 \end{cases} \tag{39}$$

where

$$\mathrm{Ri} = \frac{\left(\frac{g}{\theta_v} \frac{\partial \theta_v}{\partial z}\right)}{|\mathbf{S}|^2}.$$

Here, $\mathrm{Pr}_t = 1/3$ is a constant turbulent Prandtl number, and $\theta_v$ is the virtual potential temperature for a given specific humidity $q_t$ and specific liquid-water content $q_l$ (see, e.g., Deardorff (1980)).

## 4.3 Numerical stability

When high-order Galerkin methods are used to solve non-linear advection dominated problems, spurious Gibbs oscillations affect the solution and need to be addressed. ClimateMachine provides a set of spectral filters, cut-off filters, and artificial diffusion methods to remove these oscillations. While filters may be effective, we found that stabilizing the LES solution by





means of the SGS eddy viscosity alone is effective and robust; this is in agreement with results shown by Marras et al. (2015) and Reddy et al. (2021) in the case of continuous Galerkin methods. This approach stems from the idea that the unresolved

scales are responsible for the numerical oscillations of numerical solutions. Detailed analyses of the interactions of subgrid-scale models and filtering techniques with DG numerics in the context of atmospheric flows will be presented in a forthcoming paper.

## 5 Numerical experiments and discussions

ClimateMachine is tested against the following set of standard benchmarks: (1) dry rising thermal bubble in a neutrally strati-

fied atmosphere; (2) dry density current; (3) hydrostatic and non-hydrostatic mountain-triggered linear gravity waves; (4) the Barbados Oceanographic and Meteorological Experiment (BOMEX); and (5) decaying Taylor-Green vortex in a triply periodic domain.

All tests are executed in a 3D domain even when the problem is effectively two dimensional, with effectively zero tendencies in the third dimension; this setup is identified as 2.5D in what follows.

## 5.1 2.5D Rising thermal bubble in a neutrally stratified atmosphere

A neutrally stratified atmosphere with uniform background potential temperature $\theta_0 = 300$ K is perturbed by a circular bubble of warmer air. The hydrostatic background pressure decreases with $z$ as

$$p = p_0 \left( 1 - \frac{g}{c_{pd}\theta_0} z \right)^{c_{pd}/R_d} \tag{40}$$

in a domain $\Omega = [0, 10000] \times [-\infty, \infty] \times [0, 10000]$ m$^3$. The perturbation is as defined in Ahmad and Lindeman (2007),

$$\Delta\theta = \theta_c \left[ 1.0 - \frac{r}{r_0} \right] \qquad \text{if } r \leq r_0 = 2000 \text{ m}, \tag{41}$$

where $r = \sqrt{(x - x_c)^2 + (z - z_c)^2}$, $(x_c, z_c) = (5000, 2000)$ m, and $\theta_c = 2$ K. The initial velocity field is zero everywhere. Periodic boundary conditions are used along $y$, and solid walls with impenetrable, free-slip boundary conditions are used in the $x$ and $z$ directions. Detailed information on boundary conditions for all test cases is provided in Appendix A. Five runs are performed at effective uniform resolutions $\Delta x = \Delta z = 250$ m, 125 m, 62.5 m, and 31.25 m, and 15.625 m, with polynomial

order $N = 4$. Potential temperature $\theta$ and the two velocity components $u$ and $w$ are plotted at $t = 1000$ s in Figure 1 for the grid resolution of 15.625 m, which represents a reference solution for comparison with solutions at coarser resolutions. The value of the maximum potential temperature perturbation $\Delta\theta_{\max}$ and of the horizontal and vertical velocity components agree with the 125 m resolution results shown by Ahmad and Lindeman (2007). The grid dependence of the solution is shown in Figure 2, where potential temperature is plotted for $\Delta x = \Delta z = 31.25$ m, 62.5 m, 125 m, and 250 m. While the solution is visibly more

dissipative at coarser resolutions, the bubble's leading edge position (and hence propagation speed) is not sensitive to the grid resolution.





The SL and Vreman closures are used to model diffusive fluxes in this problem. The solutions show no discernible differences, and only the SL solution is shown. A visual comparison of the two becomes more meaningful when shear triggers mixing, which is shown for the density current test in Section 5.2. Although DG inherits an implicit numerical diffusion as an

effect of the numerical flux calculation across elements, under-resolved advection-dominated problems still require a dissipation or filtering mechanism to preserve the solution's stability. In the case of Marras et al. (2015), a dynamically adaptive SGS model was used whereas a Boyd-Vandeven filter (Boyd, 1996; Vandeven, 1991) was used by Giraldo and Restelli (2008).

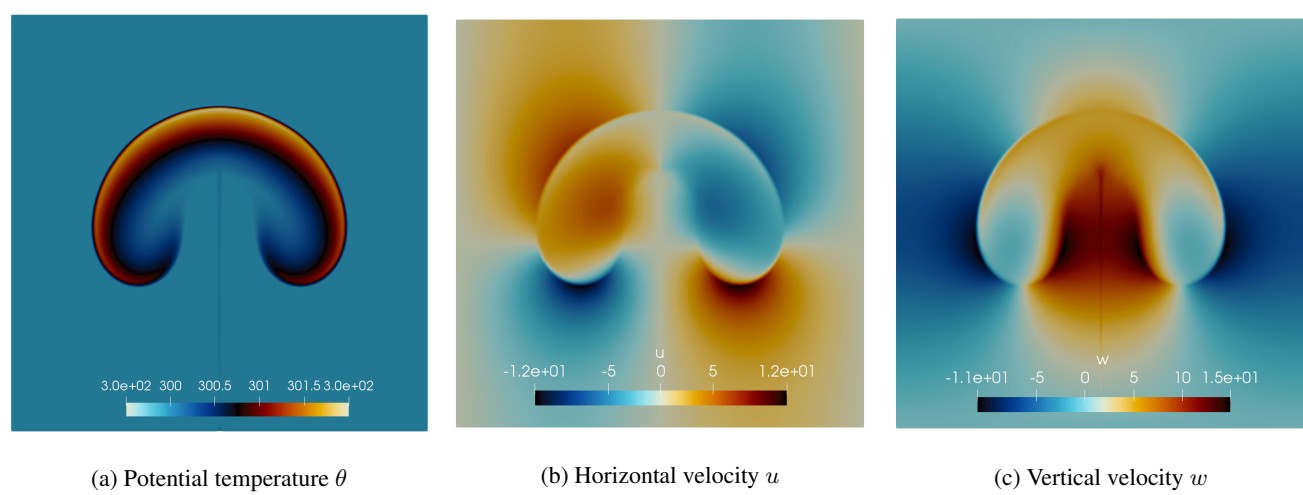

(a) Potential temperature $\theta$       (b) Horizontal velocity $u$       (c) Vertical velocity $w$

**Figure 1.** 2.5D rising thermal bubble with effective resolution $\Delta x = \Delta z = 15.625$ m and $N = 4$. Panels depict (a) potential temperature $\theta$, (b) horizontal velocity $u$, and (c) vertical velocity $w$ at t=1000 s in $\Omega = 10 \times 10$ km$^2$.

### 5.2   2.5D Density current in a neutrally stratified atmosphere

The density current problem by Straka et al. (1993) is used to test the LES framework in a flow with Kelvin-Helmholtz

instabilities. As for the rising thermal bubble, the background initial state is in hydrostatic equilibrium at uniform potential temperature $\theta_0 = 300$ K. A perturbation of $\theta$ centered on $(x_c, z_c) = (0, 3000)$ m and with radii $(r_x, r_z) = (4000, 2000)$ m is given by the function

$$\Delta\theta = \frac{\theta_c}{2}\left[1 + \cos(\pi_c r)\right] \qquad \text{if } r \leq 1 \tag{42}$$

where $\theta_c = -15$ K and $r = \sqrt{(x - x_c)/r_x^2 + (z - z_c)/r_z^2}$ in the domain $\Omega = [0, 25600] \times [-\infty, \infty] \times [0, 6400]$ m$^3$. Periodic

boundary conditions are used along $y$; impenetrable free-slip conditions are imposed in $x$ and $z$. The flow is initially stationary.

To reach solution grid convergence, this test is classically executed with a constant kinematic viscosity $\nu = 75$ m$^2$ s$^{-1}$. Increasingly finer structures are resolved when the resolution increases (Marras et al., 2012, 2015). A measure of solution fidelity is the front position, which we compare against other models in Table A1 for different resolutions. The ClimateMachine results show quantitative agreement with respect to the frontal location from a range of models with varying spatial discretizations

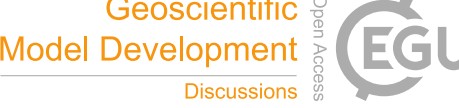

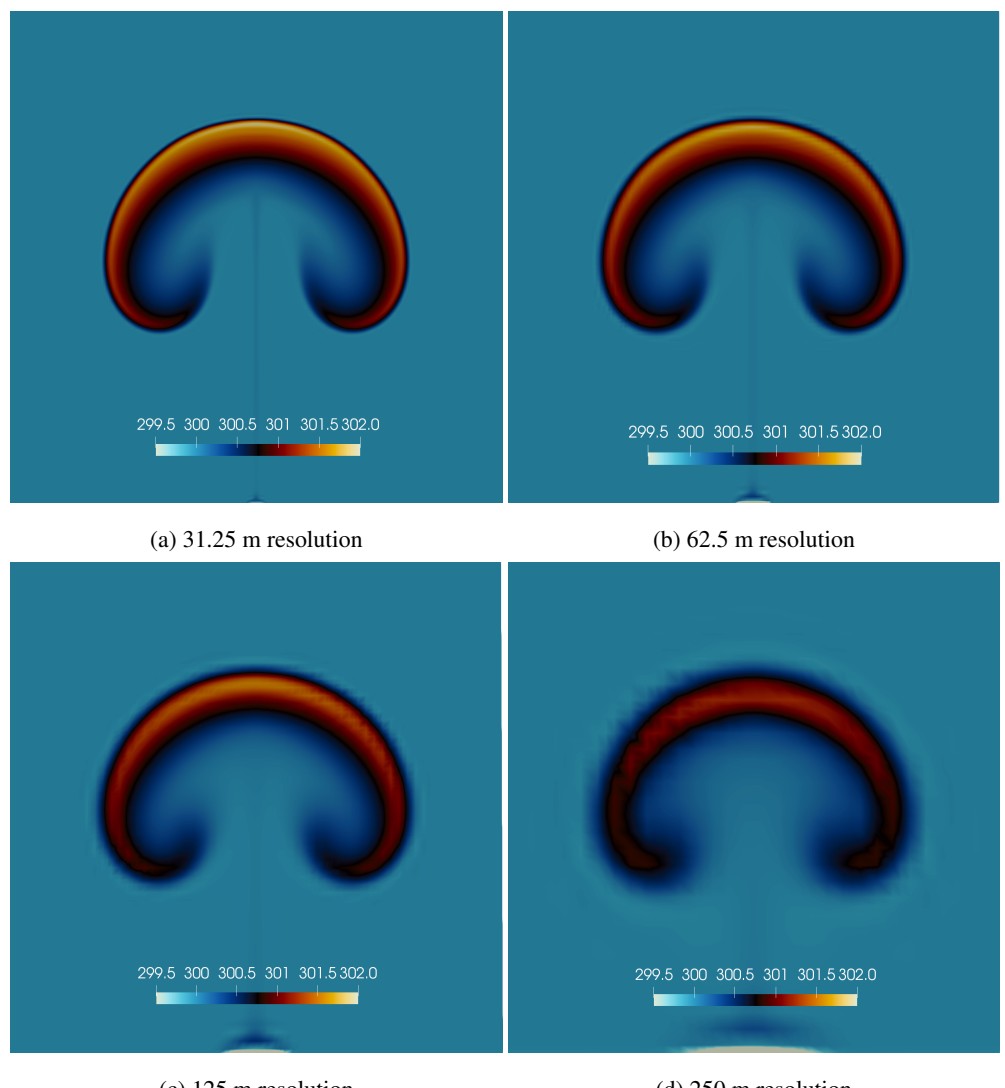

**Figure 2.** 2.5D rising thermal bubble solution with $N = 4$ at decreasing effective resolution. Grid convergence of potential temperature $\theta$ at four different resolutions to be compared against the 15.625 m resolution results shown in Figure 1. From left to right: $\Delta x = \Delta z = 31.25$ m, 62.5 m, 125 m, and 250 m. Although the solution is visibly more dissipated at coarser resolution, the bubble's leading edge (and hence propagation speed) is not affected.

at resolutions ranging from 12.5 m to 100 m. This demonstrates the scope for capturing small-scale flow features with the numerics described in Section 3.

The structure of potential temperature at the final time $t = 900$ s is shown in Figure 3 for the Vreman and SL solutions. When Rusanov is the chosen numerical flux (Figures 3a and 3b), the solutions are very similar although Vreman is visibly less dissipative for a prescribed value of the Smagorinsky coefficient $C_s = 0.18$, with finer scales of motion apparent in the





contours of potential temperature. Further quantitative analysis of the dissipative properties of the SGS models is reported in
Section 5.5. Despite Vreman being less dissipative than SL, the Roe (Figure 3c) and HLLC (Figure 3d) fluxes contribute to
additional numerical diffusion when compared to Rusanov fluxes. Detailed analysis of the interaction between numerical fluxes
and subgrid-scale models will be presented in future articles.

### 5.3   Passive transport over warped grids

To verify the correct behavior of the DG implementation in the presence of topographic features, the simple passive advection
test described by Schär et al. (2002) is used. The conservation law for the diffusive transport of a passive tracer $\chi$ is

$$\frac{\partial(\rho\chi)}{\partial t} + \nabla \cdot (\rho\boldsymbol{u}\chi) = -\nabla \cdot (\rho\boldsymbol{d}_\chi), \tag{43}$$

which is approximated via DG in the same way as Eq. (16). For a scalar tracer variable $\chi$, we model diffusive fluxes $\boldsymbol{d}_\chi$ such
that

$$\boldsymbol{d}_\chi = -(\delta_\chi \boldsymbol{\mathcal{D}}_t \boldsymbol{\nabla}\chi), \tag{44}$$

where $\delta_\chi$ relates the ratio of turbulent diffusivity of the tracer to that of the energy and moisture variables. For this test case,
however, tracer diffusivity is set to zero to assess the stability and transport properties when using a warped grid. The volume
grid in ClimateMachine is built by stacking elements above the surface and warping them around the terrain profile. To reduce
the element distortion across the domain, a linear grid damping function is used such that a topography conforming surface
of nodal points near the domain's bottom surface decays to a horizontal plane at higher altitudes (Gal-Chen and Somerville,
1975).

The initial scalar field $\chi$ is described by an elliptical perturbation centered on $(x_c, z_c) = (25, 9)$ km, with radii $(r_x, r_z) = (25, 3)$ km, such that

$$\chi = \begin{cases} \chi_0 \cos^2\left(\frac{\pi r}{2}\right), & \text{for } r \leq 1, \\ 0 \text{ otherwise} \end{cases} \tag{45}$$

where $\chi_0 = 1$ and $r = \sqrt{(x - x_c)/r_x^2 + (z - z_c)/r_z^2}$ in the domain $\Omega = [0, 150000] \times [-\infty, \infty] \times [0, 30000]$ m$^3$. An effective
uniform grid resolution $\Delta x = \Delta z = 500$ m is used for this test. The initial velocity profile is given by

$$u(z) = u_0 \begin{cases} 1, & \text{for } z \geq z_2 \\ \sin\left(\frac{\pi}{2}\frac{z - z_1}{z_2 - z_1}\right), & \text{for } z_1 \leq z < z_2, \\ 0, & \text{for } z < z_1, \end{cases} \tag{46}$$

where $u_0 = 10$ m s$^{-1}$, $z_1 = 4$ km, and $z_2 = 5$ km.



**Figure 3.** 2.5D density current. Potential temperature $\theta$ (K) at $t = 900$ s computed with an effective DG resolution of $\Delta x = \Delta z = 12.5$ m with domain extents shown in meters. (a) Rusanov numerical flux with SL SGS. (b) Rusanov numerical flux with Vreman SGS. (c) Roe numerical flux with Vreman SGS. (d) HLLC numerical flux with Vreman SGS. The color scale ranges from $\theta = 285$ to 300 K. Shared colorbar for all plots shown in panel (a) for clarity.





The topography is defined by the function

$z_{\mathrm{sfc}}(x) =$

$$
\begin{cases}
h_0 \cos^2\left(\frac{\pi(x-x_0)}{2a}\right)\cos^2\left(\frac{\pi(x-x_0)}{\lambda}\right) & \text{for } |x-x_0| \le a \\
0, & \text{for } |x-x_0| > a,
\end{cases}
\tag{47}
$$

where $h_0 = 3$ km, $a = 25$ km, $\lambda = 8$ km, and $x_0 = 75$ km.

The contours of $\chi$ in Figure 4 show minimal distortion in spite of the warped elements directly above the topographical feature, indicating that the DG transformation metrics from physical to logical space in the presence of topography do not adversely affect the solution. Deviation from the initial profile of tracer magnitudes lie between -4% and +2% when the tracer is above the topographical feature, with maximum deviation amplitudes of -5% and +3% at the end of the test, showing a favorable comparison with the hybrid and SLEVE coordinate results presented in Schär et al. (2002).

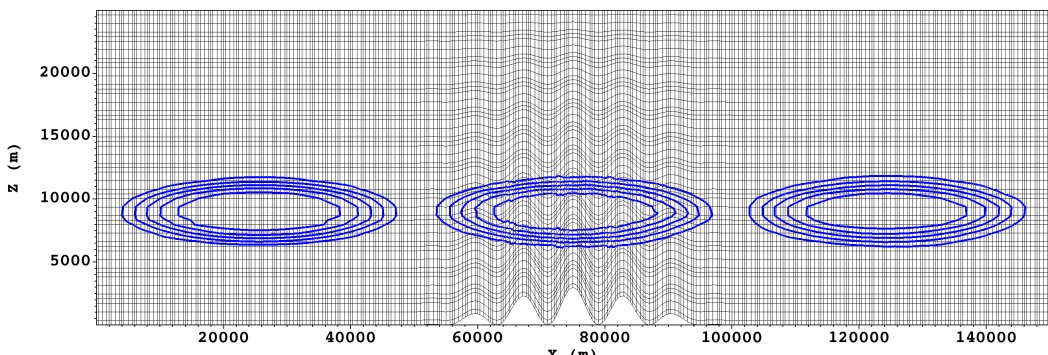

**Figure 4.** Solution of the passive transport of scalar $\chi$ at three different time instances, with contours of tracer quantity $\chi$ (from 0, outermost contour, to 1, innermost contour) overlaid on a representation of nodal points on the underlying mesh. Tracer advection is driven by a prescribed velocity profile from left to right. As it crosses the deformed grid above the mountain ridge, only a minimal distortion of tracer contours is observed, which is completely recovered back to a smooth solution downwind of the ridge.

## 5.4 Mountain-triggered gravity waves

To assess the correct implementation of a Rayleigh sponge layer to attenuate fast, upward propagating gravity waves before they reach the top of the domain, two steady-state mountain-triggered gravity wave problems suggested by Smith (1980) are solved. The sponge layer is described in Appendix A2. These problems consist of a flow that moves eastward with uniform horizontal velocity $\boldsymbol{u} = (u,0,0)$ m s$^{-1}$ in a doubly periodic domain. The flow impinges against a mountain of height $h_m$ and base length $a$ centered at $x_c$ as

$$
z_{\mathrm{sfc}}(x) = \frac{h_m a^2}{(x-x_c)^2 + a^2}.
\tag{48}
$$





The background state is in hydrostatic balance with Brunt-Väisälä frequency $\mathcal{N}$, such that

$$\theta = \theta_{\mathrm{sfc}} \, \exp\left(\frac{\mathcal{N}^2}{g} z\right)$$

for a given surface potential temperature $\theta_{\mathrm{sfc}} = T_{\mathrm{sfc}}$. The hydrostatically balanced pressure is

$$p = p_{\mathrm{sfc}} \left[1 + \frac{g^2}{c_{pd}\theta_{\mathrm{sfc}}\mathcal{N}^2}\left(\exp\left(\frac{-z\mathcal{N}^2}{g}\right) - 1\right)\right]^{c_{pd}/R_d} \tag{49}$$

which yields, by means of the ideal gas law, the background density

$$\rho = \frac{p_{\mathrm{sfc}}}{\theta R_d \left(\frac{p}{p_{\mathrm{sfc}}}\right)^{c_{vd}/c_{pd}}}. \tag{50}$$

These tests are affected by spurious oscillations that appear at approximately $5000$ s into the simulation. In the absence of shear, because of the free-slip bottom and top boundaries, the SGS models are unable to introduce sufficient diffusion to remove the Gibbs modes so that an exponential filter (Hesthaven and Warburton, 2008b) of order 64 was applied on the velocity field

to remove spurious modes. The filter assumes the form of

$$\sigma(\eta) = e^{-\alpha\eta^s}, \tag{51}$$

where $s$ is the filter order, $\eta$ is a function of the polynomial order, and $\alpha = -\log(\varepsilon_M)$ is a parameter that controls the smallest value of the filter function for machine precision $\varepsilon_M$. In double precision, $\alpha \approx 36$. The filter in this form is applied to perturbations of the prognostic variables from the balanced background state.

### 5.4.1 Linear hydrostatic


The linear hydrostatic case proposed by Smith (1979) consists of a neutrally stratified isothermal atmosphere with $\theta = \theta_{\mathrm{sfc}} = 250$ K. The background atmosphere is isothermal with temperature $T_0$, resulting in a Brunt-Väisälä frequency of

$$\mathcal{N} = \frac{g}{\sqrt{c_{pd}T_0}}.$$

The flow moves in a periodic channel along the $x$-direction with velocity $\boldsymbol{u} = (20 \text{ m s}^{-1}, 0, 0)$ over a mountain with $h_m = 1$ m

and $a = 10000$ m. A Rayleigh absorbing layer is added at $z_{\mathrm{s}} = 25$ km with relaxation coefficient $\alpha = 0.5$ s$^{-1}$, power $\gamma = 2$ and domain top $z_{\mathrm{top}} = 30$ km (see Appendix A2 for details). The domain extends from 0 to 240 km in the horizontal direction.

The steady-state solution at $t = 15000$ s is shown in Figure 5a. It is consistent with the DG results shown by Giraldo and Restelli (2008).

### 5.4.2 Linear non-hydrostatic

The linear non-hydrostatic mountain waves are forced by a flow of uniform horizontal velocity $\boldsymbol{u} = (10 \text{ m s}^{-1}, 0, 0)$ over a mountain with $h_m = 1$ m and $a = 1000$ m. The domain extends from 0 to 144 km in the horizontal direction and is 30 km high.





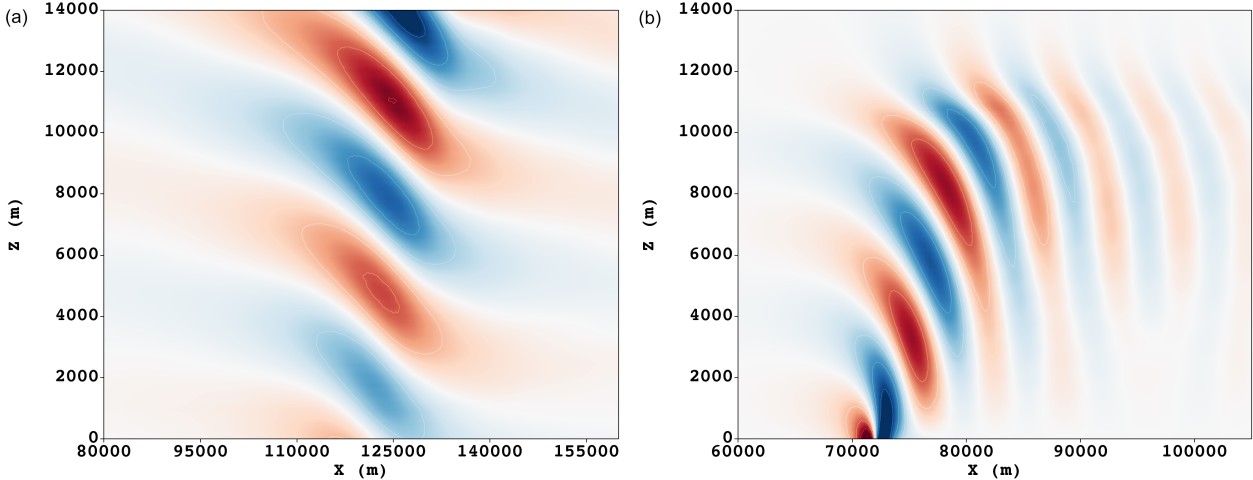

**Figure 5.** Vertical velocity $w$ for two linear mountain wave tests. Velocity contours are shown in the range from $-4 \times 10^{-3}$ m s$^{-1}$ (blue) to $4 \times 10^{-3}$ m s$^{-1}$ (red). (a) Hydrostatic test at $t = 15000$ s, with an effective horizontal resolution $\Delta x = 500$ m and effective vertical resolution $\Delta z = 240$ m with $N = 4$. (b) Non-hydrostatic test at $t = 18000$ s, with an effective horizontal resolution $\Delta x = 200$ m and effective vertical resolution $\Delta z = 160$ m with $N = 4$.

The steady-state solution at $t = 18000$ s is shown in Figure 5b. It is consistent with results shown by Giraldo and Restelli (2008).

## 5.5 Decaying Taylor-Green Vortex

The decaying Taylor-Green vortex (TGV) is a classical test to estimate the dissipative properties of turbulence models in the absence of solid boundaries. The gravity-free flow is initialized in a triply periodic cube of dimensions $[-\pi, \pi]^3$. The solenoidal initial velocity field $\boldsymbol{u_0} = (u_0, v_0, w_0)$ is defined as

$$u_0 = U_0 \sin(kx) \cos(ky) \cos(kz), \tag{52}$$

$$v_0 = U_0 \cos(kx) \sin(ky) \cos(kz), \tag{53}$$

$$w_0 = 0, \tag{54}$$

with initial pressure

$$p_0 = p_\infty + \frac{\rho_0 U_0^2}{16} (2 + \cos(kz))(\cos(2kx) + \cos(2ky)), \tag{55}$$

where $k$ is the wavenumber, $U_0 = 100$ m s$^{-1}$, $\rho_0 = 1.178$ kg m$^{-3}$, and $p_\infty = 101325$ Pa. Fourth-order polynomials are used for all simulations considered in this section.





We first consider the volume-averaged kinetic energy, which provides insight into the dissipation characteristics of the flow with respect to non-dimensionalized time $t^* = kU_0 t$. In integral form, the kinetic energy can be written as:

$$E_k = \frac{1}{2}\langle|\boldsymbol{u}|^2\rangle = \frac{1}{2\Omega^h}\int_{\Omega^h}\boldsymbol{u}\cdot\boldsymbol{u}\,d\Omega^h, \tag{56}$$

where $\langle\cdot\rangle$ denotes a volumetric average over the volume $\Omega^h$. If the flow is inviscid, the kinetic energy should be conserved.
This is only valid if the numerics or SGS models do not introduce numerical dissipation, or if all flow scales are well resolved. As such, the time series of kinetic energy is a metric that shows the point along the simulation at which the solution becomes under-resolved. The kinetic energy dissipation rate is the second quantity of interest, which allows us to quantify the rate of decay of kinetic energy over time. This is defined as

$$\epsilon = -\frac{dE_k}{dt}. \tag{57}$$

A third quantity of interest for this analysis is enstrophy, which is defined as the square of the vorticity norm:

$$\langle\omega^2\rangle = \langle||\nabla\times\boldsymbol{u}||^2\rangle. \tag{58}$$

The enstrophy of a fully resolved flow should go to infinity if the flow is inviscid. Therefore, enstrophy can be used as a criterion to estimate the effect of numerical dissipation.

By means of a three-dimensional fast fourier transform (FFT) of the velocity field, the kinetic energy spectrum is calculated as:

$$E(k) = \int_0^{2\pi}\int_0^{\pi}\int_0^{K} A(k_x, \upsilon, \zeta) k^2 \sin(\zeta)\, dk\, d\upsilon\, d\zeta, \tag{59}$$

where $K = 2\pi/L$, $L$ is the characteristic length, $A$ is a three dimensional array of Fourier mode amplitudes, $\upsilon = k_z/k$ and $\zeta = \tan^{-1}(k_y/k_x)$ and $k = \sqrt{k_x^2 + k_y^2 + k_z^2}$. The TGV flow is simulated using both the SL and Vreman models on grids with $32^3$, $64^3$, $128^3$, and $192^3$ points. Figure 6 shows a 3D visualization of the flow at two different non-dimensional times using zero Q-criterion isosurfaces, which identify balance between rotation and shear in the flow (Hunt et al., 1988). As the flow evolves, the flow generates smaller and smaller-scale vortices. Eventually, the flow becomes under-resolved, making it impossible to conserve kinetic energy. As the flow continues to evolve, an instability occurs, which causes the disintegration of the vortex sheet. After this point, the TGV's dynamics are controlled by the interaction of small-scale vortical structures formed by vortex stretching.

Results for the coarse-resolution simulations are presented in Figure 7, and those for the fine-resolution simulations are shown in Figure 8. Figure 7a shows that the kinetic energy changes with time for the $32^3$ resolution simulations are distinctly different from their higher-resolution counterparts in Figure 8a. The severe under-resolution of the flow seems to generate much larger amounts of dissipation early on. This is further demonstrated when comparing Figures 7b and 8b. Beyond $64^3$ resolution, the peaks in dissipation are larger as the resolution increases, since smaller vortices can be resolved before the



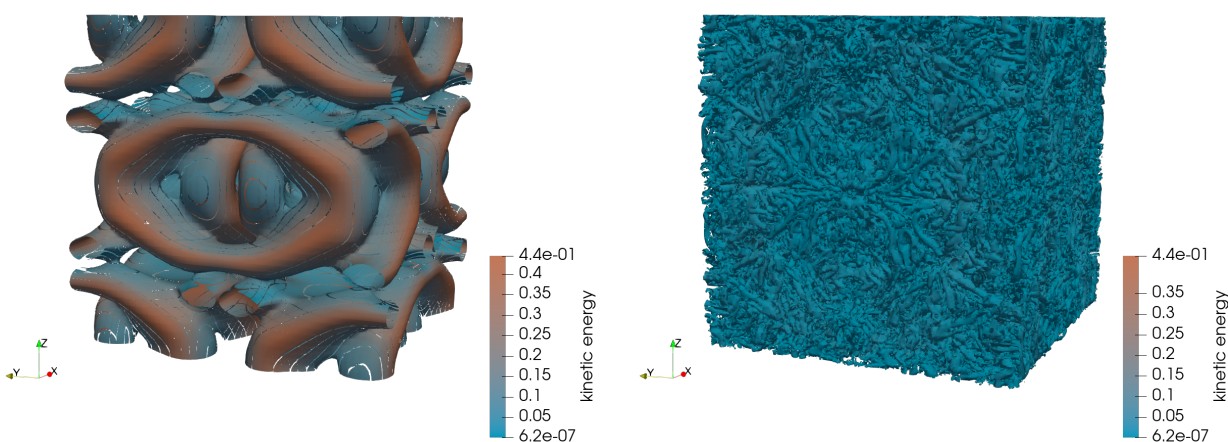

**Figure 6.** Taylor-Green Vortex. Isosurfaces of zero Q-criterion (these idenfity surfaces where the vorticity norm is identical to the strain-rate magnitude) on a $192^3$ grid, corresponding to 48 elements of order 4 at $t^* = 4$ (left) and $t^* = 60$ (right). Plots are shown for the Vreman solution. The isosurfaces of the Q-criterion are colored by dimensionless kinetic energy.

instability eventually happens. The $32^3$ simulations (Figure 7b) have larger peaks than even the $192^3$ (Figure 8b) simulations, demonstrating that the under-resolution of the vortex structures leads to different, more dissipative early-flow behavior.

Figure 8b shows that the $128^3$ and $192^3$ simulations using both SGS closures are characterized by a peak in the kinetic energy dissipation at $t^* = 9$. Brachet et al. (1983) and Brachet (1991) demonstrate this result for their direct numerical simulations (DNS) of the Taylor-Green vortex for a Reynolds number $R_e = U_0/k\nu \geq 3000$. Examining Figure 7b, the $64^3$ simulations

suggest a dissipation peak time of $t^* = 8$, and the $32^3$ simulations suggest a dissipation peak time of $t^* = 6$. The under-prediction of the time at which the peak occurs is due to an inability to resolve the vortices that appear early on in the flow's evolution at extremely coarse resolutions. This leads to the early appearance of the instability which causes the dissipation peak. Furthermore, we see that the kinetic energy decay occurs sooner for the lower-resolution simulations, as a result of increased dissipation from the SGS models.

Figure 8c also shows that the flow's enstrophy behaves as expected, with peak values at $t^* = 9$ coinciding with peaks in kinetic energy. The higher-resolution simulations are able to reach a higher enstrophy than the lower-resolution simulations as they are naturally able to resolve more vortical motion. On the other hand, the choice of SL or Vreman models seems to have very little impact on the ability to resolve more small-scale eddies for low resolutions. However, the SL SGS scheme leads to higher enstrophy than the Vreman SGS scheme, increasingly so as resolution increases.

Figure 9 shows the kinetic energy spectra obtained for the higher-resolution simulations used for this test. All simulations present peaks in their respective spectra at $k = 2$ to $4$, which persist even as the flow evolves over time. These peaks are explained by Drikakis et al. (2007) as being imprinted on the spectra by the initial velocity field. Furthermore, as the flow



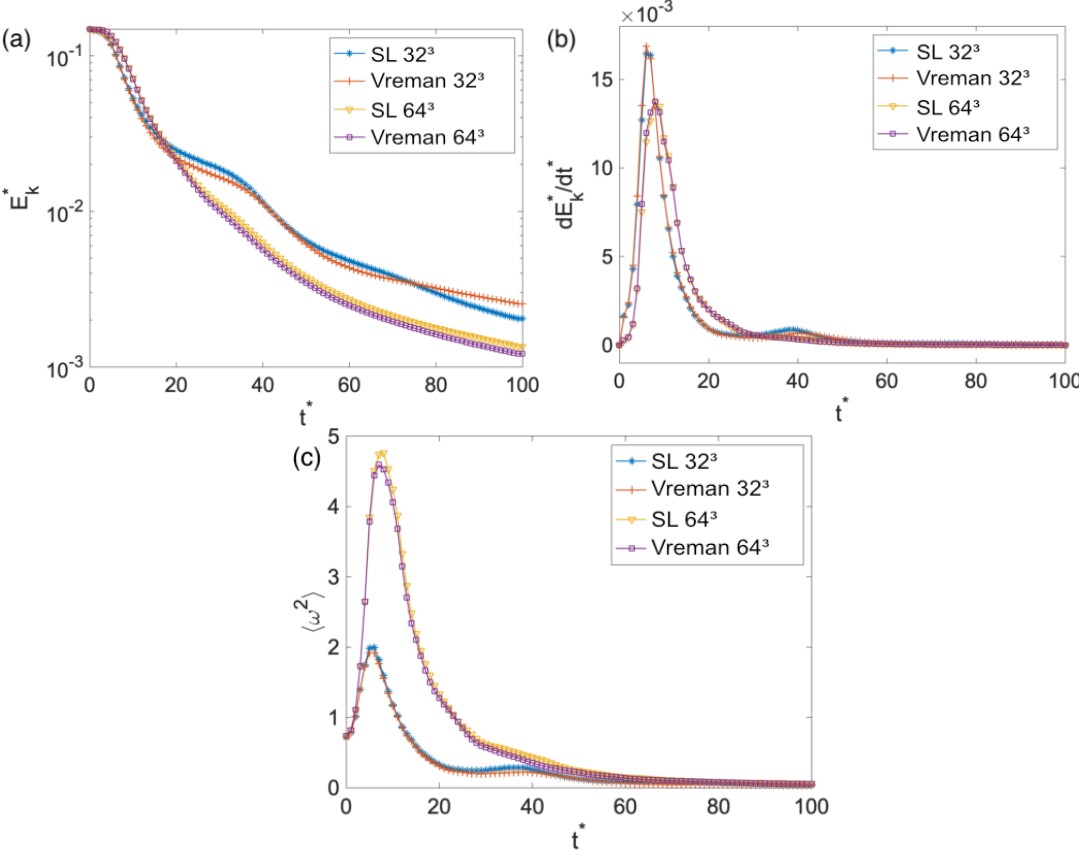

**Figure 7.** Evolution of volumetrically averaged (a) kinetic energy represented on a semi-logarithmic scale, (b) kinetic energy dissipation, and (c) enstrophy, each computed on $32^3$ and $64^3$ grids with the Vreman and SL SGS models.

evolves, all spectra show a slope close to the theoretical $k^{-5/3}$ of homogeneous turbulence and eventually decay towards a $k^{-3}$ slope at higher wavenumbers. This behavior is consistent with the DNS results of Brachet et al. (1983) who showed, using DNS, this transition occurs around $k$=60.

### 5.6 Barbados Oceanographic and Meteorological Experiment (BOMEX)

BOMEX features a shallow cumulus topped boundary layer as described in Holland and Rasmusson (1973). The setup of this test follows Siebesma et al. (2003). The initial profiles are characterized by a well-mixed sub-cloud layer below 500 m, a cumulus layer between 500 m and 1500 m, an inversion layer up to 2000 m, and a free troposphere above. Large-scale forcing includes prescribed large-scale subsidence, horizontal advective drying, radiative cooling, and Coriolis acceleration. Sensible and latent heat fluxes at the surface are prescribed to $\mathrm{SHF} = \boldsymbol{n} \cdot (\rho \boldsymbol{J}_{\mathrm{sfc}}) = 9.5\,\mathrm{W\,m^{-2}}$ and $\mathrm{LHF} = \boldsymbol{n} \cdot (\rho \boldsymbol{D}_{\mathrm{sfc}}) = 147.2\,\mathrm{W\,m^{-2}}$. Additional detail on the application of boundary conditions is presented in Appendix A. The domain, $\Omega = 6400 \times 6400 \times 3000\,\mathrm{m^3}$, is doubly periodic in the $x$ and $y$ directions. A Rayleigh sponge layer (see Appendix A2 for details) is applied along





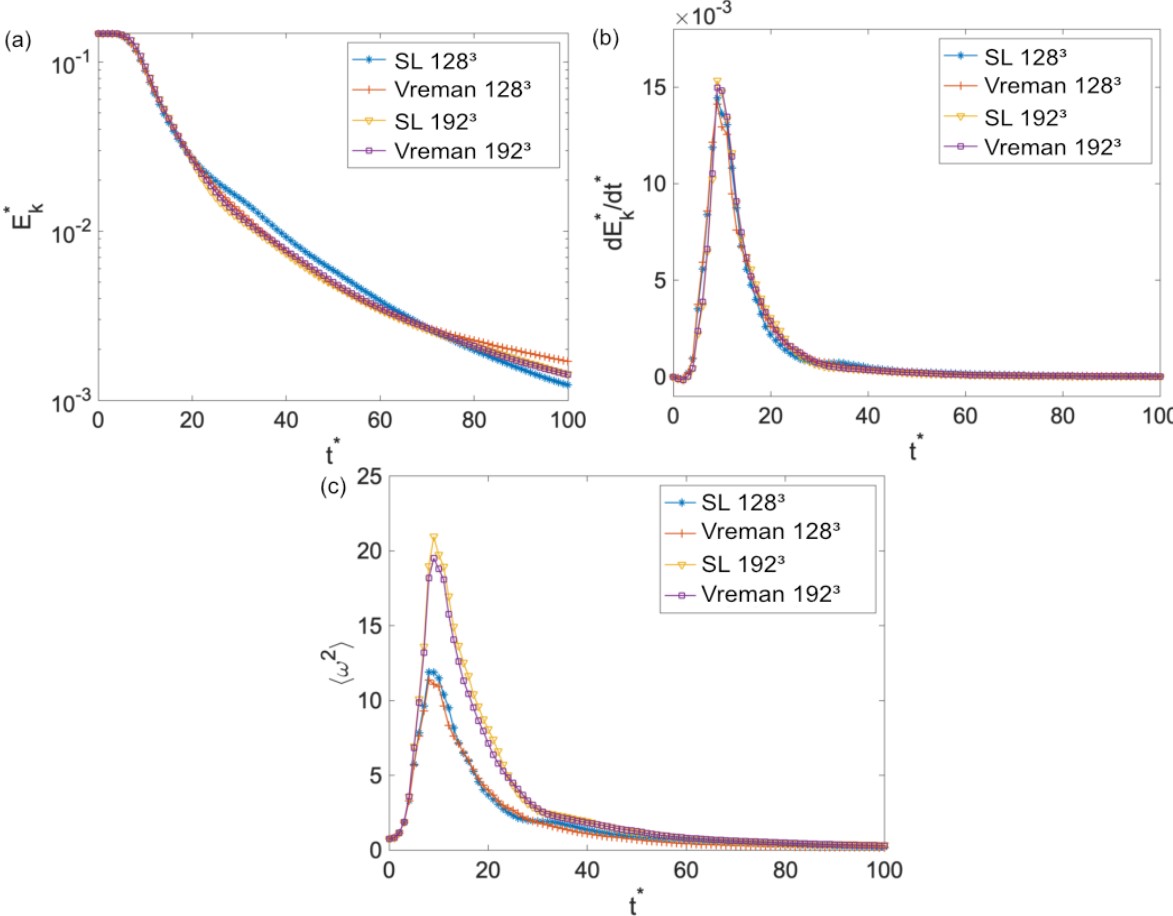

**Figure 8.** Evolution of volumetrically averaged (a) kinetic energy represented on a semi-logarithmic scale, (b) kinetic energy dissipation and (c) enstrophy, each computed on $128^3$ and $192^3$ grids with the Vreman and SL SGS models.

the $z$ direction to damp upward propagating gravity waves. On the bottom surface, a momentum drag forcing is applied (see

Appendix A1). The effective horizontal and vertical resolutions are, respectively, $\Delta x = \Delta y = 100$ m and $\Delta z = 40$ m. The simulation time is 6 hours.

Figure 10 shows the vertical profiles of the domain-mean thermodynamic and turbulence properties over the last hour of the simulations. Figure 11 shows the time series of liquid water paths (LWP), cloud cover, and turbulence kinetic energy. The time averaged results of the vertical profiles of $\theta_l, q_l$, cloud fraction, and $q_v = q_t - q_l$ during the last hour are in good agreement

with the same quantities presented by Siebesma et al. (2003). The SGS model does not have much effect on the simulation characteristics, except that the Vreman SGS model produces a stronger peak in the variance of vertical velocity, $\overline{w'w'}$, near the cloud top. Although the difference is mild, it is possibly due to the low dissipation nature of the Vreman's model. Excluding the first hour of flow spin up, the results compare well with PyCLES (Pressel et al., 2015) and fall within the ensemble range





**Figure 9.** Kinetic energy spectra obtained using (a) $128^3$ points and SL, (b) $128^3$ points and Vreman, (c) $192^3$ points and SL, and (d) $192^3$ points and Vreman.

shown in Figure 2 of Siebesma et al. (2003). Details on the computation of horizontally-averaged profiles can be found in
Appendix B.

A large domain simulation of BOMEX with effective horizontal resolution $\Delta x = \Delta y = 50$ m and vertical resolution $\Delta z = 20$ m in a $30 \times 30 \times 3.6$ km$^3$ domain was executed using 16 GPUs on the Google Cloud Platform. The simulation was executed using 1D IMEX time integration (1D implicit in the vertical direction and 2D explicit in the horizontal direction) at maximum horizontal advective Courant number C = 0.9. A visualization of instantaneous shallow cumulus structures is shown
in Figure 12.



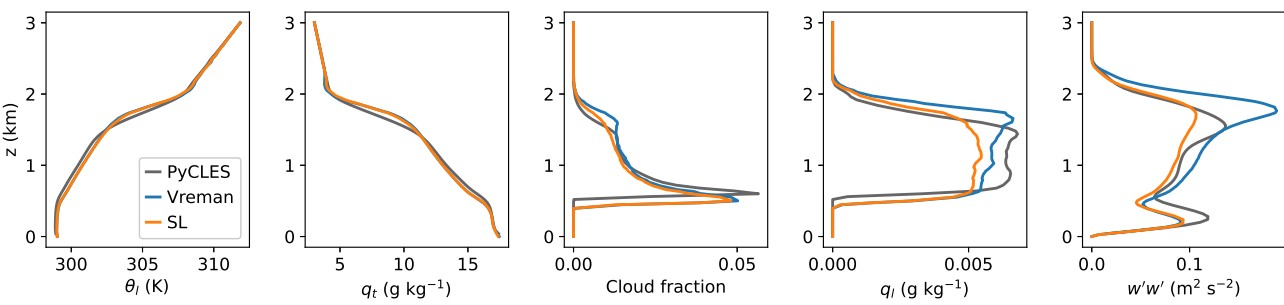

**Figure 10.** BOMEX. Profile of the mean state of liquid potential temperature, total specific humidity, cloud fraction, liquid water specific humidity, and variance of the vertical velocity fluctuations averaged along the last hour of the simulation. The solutions with PyCLES and ClimateMachine were calculated with effective grid resolution $\Delta x = \Delta y = 100$ m and $\Delta z = 40$ m. Results calculated with Vreman (blue lines) and SL SGS (orange lines) models in ClimateMachine are compared against results of PyCLES (grey lines) in its Paired-SGS modality using the SL model.

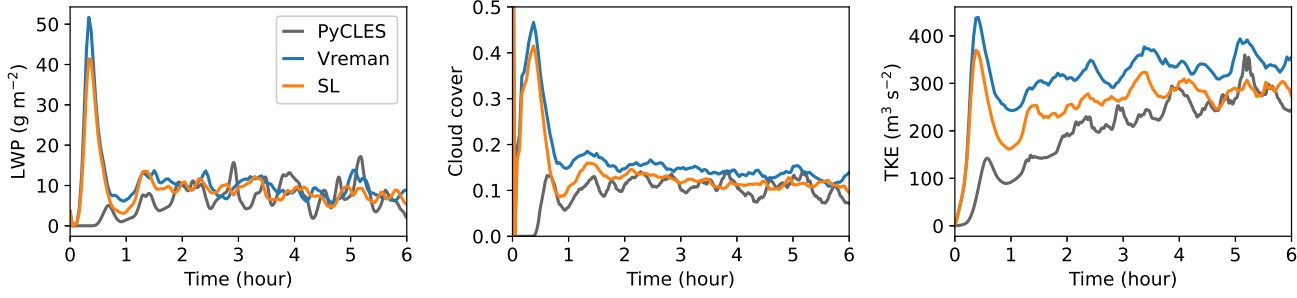

**Figure 11.** BOMEX. From left to right, time series of horizontally averaged LWP, cloud cover, and turbulence kinetic energy diagnosed from ClimateMachine using Vreman and SL. These results are consistent with ensemble results presented in Figure 2 of the intercomparison study by Siebesma et al. (2003). Line colors as in Figure 10.

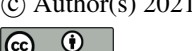


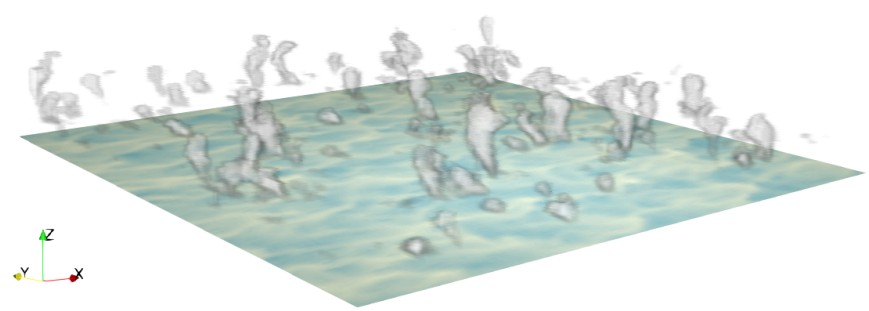

**Figure 12.** BOMEX. Instantaneous visualization of the shallow cumulus structures on a $30\,\text{km} \times 30\,\text{km} \times 3.6\,\text{km}$ domain. The white shading of the volume rendering of the cloud corresponds to a maximum value of $q_l = 2.5 \times 10^{-4}\,\text{kg kg}^{-1}$; a maximum $q_t = 1.8 \times 10^{-2}\,\text{kg kg}^{-1}$ is shown on the bottom surface in light yellow shading. The simulation was executed using 16 GPUs on the Google Cloud Platform.

## 6 Mass and energy conservation

We define the time dependent normalized total mass and energy changes as, respectively,

$$\Delta M(t) = \frac{\int_\Omega \left[\rho(t) - \rho(t_0)\right] d\Omega}{\int_\Omega \rho(t_0) d\Omega}, \tag{60}$$

and

$$\Delta \rho e^{tot}(t) = \frac{\int_\Omega \left[\rho e^{tot}(t) - \rho e^{tot}(t_0)\right] d\Omega}{\int_\Omega \rho e^{tot}(t_0) d\Omega}, \tag{61}$$

where $t_0$ indicates the initial time and $\Omega$ is the full domain. Figure 13 shows $\Delta M(t)$ and $\Delta \rho e^{tot}(t)$ for a 1 hour simulation of a moist rising thermal bubble. The SL eddy viscosity model was used to represent under-resolved diffusive fluxes. This simulation was run using free-slip boundary conditions with adiabatic walls. We note that the loss of energy and mass in the system is contained to $\mathcal{O}(10^{-15})$, that is, numerical roundoff-error. This result highlights a key benefit of the general formulation of the prognostic conservation equations in flux form, which guarantees conservation properties up to source or sink contributions.

## 7 CPU strong-scaling

Demonstration of favorable scaling capabilities across multiple hardware types is critical to the utility of ClimateMachine as a competitive tool for large-eddy simulations. Toward this, we first examine strong scaling on CPU architectures. The rising thermal bubble problem described in Section 5.1 is used as the test problem, with its domain extents modified to form an $8.192\,\text{km}^3$ cube, with an effective nodal resolution of 32 m to ensure that CPU memory on a single rank is maximally loaded. For tests with multiple MPI-ranks, each rank resides on a unique node, to ensure communication overhead is appropriately represented; in practice, one would expect to use multiple ranks per node. Scaling across multiple threads is not assessed



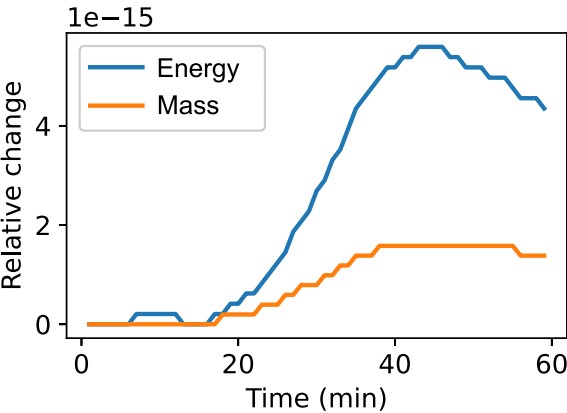

**Figure 13.** Time evolution of the mass and total energy loss (relative change when compared against initial conditions) for a moist thermal bubble simulation. Blue line: relative change in energy; Orange line: relative change in mass.

in the present work. Figure 14 shows the speedup in time-to-solution for 10 time-integration steps of the test problem with
$N_{\mathrm{ranks}} \in \{1, 2, 4, 8, 16, 32\}$ for both dry and moist simulations. We exclude checkpoint, diagnostic and periodic run-time output steps from time-to-solution measurements. In both dry and moist simulations, we see a speedup of approximately 19.7 when using 32 ranks compared with the corresponding single-rank simulation.

A single rank GPU run of the test problem on a $6.144\,\mathrm{km}^3$ domain with $32\,\mathrm{m}$ effective resolution (restricted by GPU memory capacity) has a wall-clock time for ten integration steps of $314$ s. The wall-clock time for a 32-rank CPU run was $449$ s for a
$2.37$ times larger problem.

This provides an estimate for a comparison between CPU and GPU hardware performance. However, the balance between memory bandwidth limits and compute operation limits guides the maximum scaling possible on the GPU hardware relative to its CPU counterpart, so this cannot be interpreted as a direct comparison across hardware types. Based on the present results, we conclude that it is more feasible to pursue strong-scaling improvements on CPU hardware than on GPU hardware.
Further optimization and exploration of scaling in ClimateMachine is ongoing work. Additional details on the hardware used for scaling tests can be found in Appendix C.

## 8   GPU weak-scaling

To test the multi-GPU scalability of ClimateMachine, we first execute a BOMEX setup that is sufficiently large to saturate one GPU. The single-GPU execution represents the baseline from which we calculate the average time per time-step denoted
by $t_1$. We then expand the domain size to match an increase in the number of GPUs and measure the average time per time step. Our scaling is then obtained as the ratio $t_1/t_n$, with $t_n$ being the average time per time-step obtained with $n$ GPUs. The results are obtained using up to 16 NVIDIA Tesla V100 GPUs running Julia version 1.4.2, CUDA 10.0, and CUDA-aware



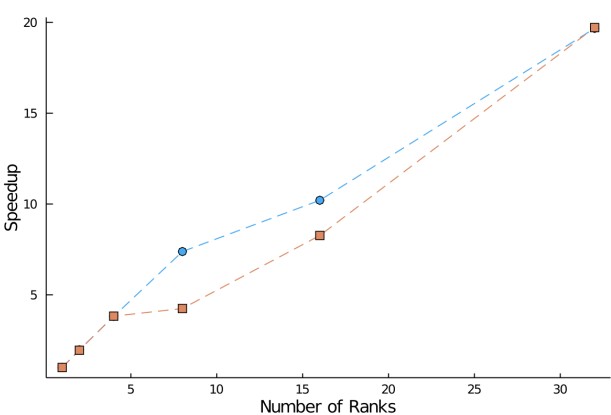

**Figure 14.** Speedup of time-integration (solver) step relative to time to solution for single-rank simulation of the rising thermal bubble problem in an $8.192$ km$^3$ domain with uniform effective resolution of 32 m (with fourth-order polynomials). Blue circles: dry simulation; Orange squares: moist simulation.

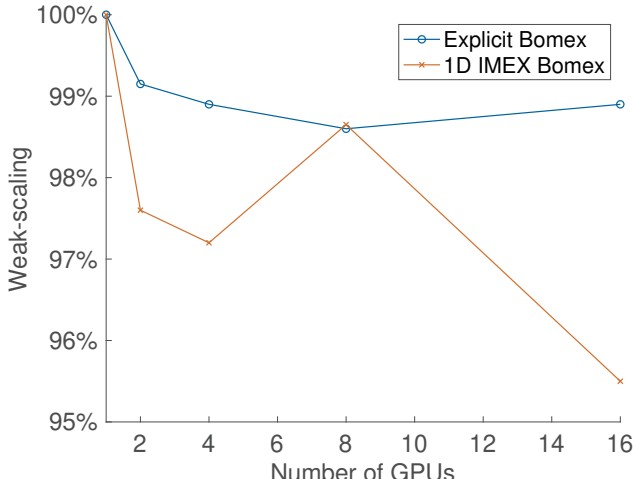

**Figure 15.** BOMEX weak scaling using 1D IMEX versus fully explicit time integration.

OpenMPI 4.0.3. Figure 15 shows excellent weak scaling for up to 16 GPUs on Google Cloud Platform resources. Over 95% weak scaling was achieved with 1D-IMEX time integration, and over 98% for the simulation with explicit timestepping. This
is an encouraging result and supports the ability to prototype smaller problem setups and deploy larger simulations in the ClimateMachine limited area configuration at identical resolutions without significantly compromising the time to solution.



## 9    Conclusions

This paper introduced and assessed the LES configuration of ClimateMachine, a new Julia language simulation framework designed for parallel CPU and GPU architectures. Notable features of this LES framework are:

- Conservative flux form model equations for mass, momentum, total energy and total moisture to ensure global conservation of dynamical variables of interest (up to non-conservative source or sink processes)

- Discontinuous-Galerkin discretization with element-wise evaluation of the approximations to volume and interface integrals resulting in reduced time-to-solution due to MPI operations

- Application of model equations to the solution of benchmark problems in typical LES codes, including atmospheric
flows in the shallow cumulus regime (BOMEX)

- Demonstration of strong-scaling on CPUs with up to 32 MPI-ranks (speed-up of 19.7 in time-to-solution), and weak scaling up to 16 GPUs (95-98%), in both dry and moist simulation configurations.

*Code availability.* ClimateMachine is an open source framework, and maintained on Github: https://github.com/CliMA/ClimateMachine.jl. Documentation for installing and running ClimateMachine is available at https://clima.github.io/ClimateMachine.jl/latest/. The version used
in this paper is v0.2.0, which can be downloaded from https://doi.org/10.5281/zenodo.5542395, or https://github.com/CliMA/ClimateMachine. jl/releases/tag/v0.2.0.

*Data availability.* The test cases presented in the paper are available in the following directory locations in the source code:

- Section 5.1: /tutorials/Atmos/risingbubble.jl

- Section 5.2: /tutorials/Atmos/densitycurrent.jl

- Section 5.3: /experiments/AtmosLES/schar_scalar_advection.jl

- Section 5.4: /experiments/AtmosLES/agnesi_hs_lin.jl and  /experiments/AtmosLES/agnesi_nh_lin.jl

- Section 5.5: /experiments/AtmosLES/taylor_green.jl

- Section 5.6: /experiments/AtmosLES/bomex_model.jl and  /experiments/AtmosLES/bomex_les.jl

Configuration instructions and methods for resolution, numerical flux schemes, boundary conditions and turbulence model specification
can be found in the code documentation at https://clima.github.io/ClimateMachine.jl/v0.2/. Runtime instructions can be accessed at https://clima.github.io/ClimateMachine.jl/v0.2/GettingStarted/RunningClimateMachine/.





## Appendix A: Boundary Conditions

### A1 Solid walls and wall fluxes

**Momentum** Rigid surfaces are considered impenetrable such that the wall-normal component of the velocity vanishes at rigid
boundaries by imposing $\boldsymbol{u} \cdot \boldsymbol{n} = 0$. The viscous sublayer is not explicitly resolved, and a momentum sink is applied to model
the effect of wall-shear stresses. While the wall-normal advective momentum flux vanishes, the wall-normal viscous or SGS
momentum flux, also known as the bulk surface stress (units of Pa),

$$\boldsymbol{n} \cdot (\rho\boldsymbol{\tau}) = -\boldsymbol{n} \cdot [2\rho\boldsymbol{\nu}_t \boldsymbol{S}(\boldsymbol{u}_p)], \tag{A1}$$

is not necessarily negligible. Here, $\boldsymbol{S}(\boldsymbol{u}_p) = (\nabla\boldsymbol{u}_p + \nabla\boldsymbol{u}_p^T)/2$ is the strain rate tensor of the near-surface wall-parallel velocity,
$\boldsymbol{u}_p$. We note that, throughout appendix A, $\boldsymbol{n}$ refers to the inward-pointing normal vector at domain boundaries, distinct from
the prior definition of the element-interface normal vector in Section 3.

In the case of free-slip conditions at a solid surface (indicated by subscript "sfc"), there is no viscous or SGS momentum
transfer between the atmosphere and the surface, such that

$$\boldsymbol{n} \cdot (\rho\boldsymbol{\tau})\big|_{\text{sfc}} = 0.$$

Because the momentum flux tensor depends linearly on velocity derivatives, this amounts to homogeneous Neumann bound-
ary conditions on velocity components parallel to the surface. On the other hand, viscous drag is imposed by the classical
aerodynamic drag law

$$\boldsymbol{n} \cdot (\rho\boldsymbol{\tau}) = -\rho C_{d,\text{int}} \|\boldsymbol{u}_{p,\text{int}}\| \boldsymbol{u}_{p,\text{int}}, \tag{A2}$$

where the quantities with subscript int are evaluated at an interior point $\boldsymbol{x}_{\text{int}}$ adjacent to the surface. The drag coefficient

$$C_{d,\text{int}} = C_d(\boldsymbol{Y}; \boldsymbol{x}_{\text{int}})$$

can depend parameterically on state variables $\boldsymbol{Y}(\boldsymbol{x}, t)$ and on the position $\boldsymbol{x}_{\text{int}}$ of the interior point relative to the surface. In
the present implementation, the plane of interior points relevant to boundary flux evaluation is interpreted as the first layer
of interior nodes in the surface-adjacent elements. The drag law boundary condition amounts to inhomogeneous Neumann
boundary conditions on velocity components parallel to the surface.

**Specific humidity** As for momentum, the advective specific humidity fluxes normal to a rigid surface vanish, but the diffusive
or SGS specific humidity fluxes normal to the surface may not vanish. Normal components of SGS fluxes of condensate, $q_l$,
are generally set to zero at boundaries

$$\boldsymbol{n} \cdot (\rho\boldsymbol{d}_{q_l})\big|_{\text{sfc}} = 0, \tag{A3}$$

implying homogeneous Neumann boundary conditions ($\boldsymbol{n} \cdot \boldsymbol{\nabla} q_l = 0$) on the condensate specific humidities. The total SGS
specific humidity flux then reduces to the vapor flux at the surface. SGS turbulent deposition of condensate (fog) on the surface




can in principle occur; representing this would require nonzero condensate fluxes at the surface. With the assumption of zero condensate boundary fluxes, we have

$$\boldsymbol{n} \cdot (\rho \boldsymbol{d}_{q_t}) \big|_{\text{sfc}} = \boldsymbol{n} \cdot (\rho \boldsymbol{d}_{q_v}) \big|_{\text{sfc}},$$

where evaporation (measured in $\text{kg m}^{-2} \text{ s}^{-1}$), or condensation if negative, is given by

$\quad E = \boldsymbol{n} \cdot (\rho \boldsymbol{d}_{q_v}) \big|_{\text{sfc}} = -\boldsymbol{n} \cdot (\rho \boldsymbol{\mathcal{D}}_t \boldsymbol{\nabla} q_t) \big|_{\text{sfc}}.$

Evaporation can be zero (water impermeable) or it can be given as a function of $\boldsymbol{Y}$ at the surface according to

$$E = \boldsymbol{n} \cdot (\rho \boldsymbol{d}_{q_v}) \big|_{\text{sfc}} = E(\boldsymbol{Y}; \boldsymbol{x}_{\text{sfc}}, t),$$

which, numerically, translates into an inhomogeneous Neumann boundary condition on the vapor specific humidity.

**Energy** As for momentum and humidity, the advective energy fluxes normal to a rigid surface vanish, but the diffusive or
$\quad$ SGS flux of total enthalpy, $h^{\text{tot}}$, normal to the surface (units of $\text{W m}^{-2}$)

$$\boldsymbol{n} \cdot \rho(\boldsymbol{J} + \boldsymbol{D}) = -\boldsymbol{n} \cdot (\rho \boldsymbol{\mathcal{D}_t} \boldsymbol{\nabla} h^{\text{tot}}),$$

may not vanish. Because the kinetic energy contribution to the total enthalpy flux near a surface is usually 3–4 orders of magnitude smaller than the thermal and potential energy components, it is generally neglected, so that the total enthalpy flux $\boldsymbol{J} + \boldsymbol{D}$ reduces to a flux of moist static energy $\text{MSE} = h + \Phi$. The surface can be insulating, in which case the SGS transfer of
$\quad$ total enthalpy between the atmosphere and the surface is zero:

$$\boldsymbol{n} \cdot \rho(\boldsymbol{J} + \boldsymbol{D})_{\text{sfc}} = 0,$$

which, from a numerical point of view, translates to a homogeneous Neumann condition on the total enthalpy ($\boldsymbol{n} \cdot \boldsymbol{\nabla} h^{\text{tot}} = 0$) or, by neglecting kinetic energy, on MSE such that ($\boldsymbol{n} \cdot \boldsymbol{\nabla} \text{MSE} = 0$). If MSE is a known function, the total energy flux is given by

$\quad \boldsymbol{n} \cdot \rho(\boldsymbol{J} + \boldsymbol{D})_{\text{sfc}} = \text{MSE}(\boldsymbol{Y}; \boldsymbol{x}_{\text{sfc}}, t).$

The value of $\rho \boldsymbol{n} \cdot (\boldsymbol{J} + \boldsymbol{D})$ can also be assigned by the summation of given LHF and SHF, as done in the case of BOMEX described in Section 5.

## A2 Non-reflecting top boundary

To prevent the reflection of fast, upward propagating gravity waves at the top boundary, a Rayleigh-damping sponge is added
$\quad$ to the right-hand side of the momentum equation (see Section 5). The damping in the momentum equations takes the form:

$$\frac{\partial \rho \boldsymbol{u}}{\partial t} = \cdots - \tau_s \rho(\boldsymbol{u} - \boldsymbol{u}_{\text{relax}}) \tag{A4}$$



where $\boldsymbol{u}_{\mathrm{relax}}$ is a specified background velocity to which the flow is relaxed within the absorbing layer with a characteristic relaxation coefficient $\tau_s$. Of the many alternative options known for $\tau_s$ (e.g., Durran and Klemp (1983)), the default in ClimateMachine is

$$\tau_s = \alpha \sin^\gamma \left( \frac{1}{2} \frac{z - z_{\mathrm{s}}}{z_{\mathrm{top}} - z_{\mathrm{s}}} \right) \qquad \text{for } z > z_{\mathrm{s}} \tag{A5}$$

where the absorbing sponge layer starts at $z = z_{\mathrm{s}}$, $\gamma$ is a positive even power, typically set to 2, and $\alpha > 0$ is a relaxation coefficient, typically of order $\mathcal{O}(1\ \mathrm{s}^{-1})$.

### A3 Numerical implementation

For the boundary velocity corresponding to the impenetrable wall condition, we use the following reflecting condition

$$\boldsymbol{u}_{bc} = \boldsymbol{u}^- - (\boldsymbol{n} \cdot \boldsymbol{u}^- \boldsymbol{n}), \tag{A6}$$

$$\boldsymbol{u}^+ = 2\boldsymbol{u}_{bc} - \boldsymbol{u}^-. \tag{A7}$$

The no-slip condition follows from (A6) by setting all components of $\boldsymbol{u}_{bc} = 0$. Boundary conditions on a scalar $\chi$ are similarly specified as follows

$$\chi^+ = 2\chi_{bc} - \chi^-. \tag{A8}$$

Non-zero mass-flux boundary conditions can be imposed at penetrable or free surfaces by applying the transmissive boundary condition

$$\boldsymbol{u}^+ = \boldsymbol{u}^-. \tag{A9}$$

Diffusive fluxes are applied by a direct specification of the wall-normal fluxes, and over-specified boundary conditions are avoided by using only the interior ($^-$) gradients and first-order fluxes.

### Appendix B: Supplementary Results

This section provides additional information on the comparison of the density current benchmark in ClimateMachine with existing literature references in Table A1.





**Table A1.** Summary of frontal locations for the density current test case from existing literature. Results tabulated are of the front location at $t = 900$ s. The results are reported for the following models: Climate Machine with SL and Vreman, FEM VMS, $f$-wave, filtered Spectral Elements (SE), filtered Discontinuous Galerkin (DG), and PPM.

| Model | Space discr. | Resolution | Order | $\mu = 75 \text{ m}^2 \text{ s}^{-1}$ | Front Location [m] |
|---|---|---|---|---|---|
| ClimateMachine, SL | DG | 12.5 m | $4^{th}$ | No | 15090 |
| " | " | 25 m | $4^{th}$ | No | 14990 |
| " | " | 50 m | $4^{th}$ | No | 14770 |
| " | " | 100 m | $4^{th}$ | No | 14669 |
| ClimateMachine, Vreman | " | 12.5 m | $4^{th}$ | No | 15091 |
| " | " | 25 m | $4^{th}$ | No | 14950 |
| " | " | 50 m | $4^{th}$ | No | 14739 |
| " | " | 100 m | $4^{th}$ | No | 14606 |
| Giraldo-Restelli (Giraldo and Restelli, 2008) | DG | 50 m | $4^{th}$ | Yes | 14767 |
| Giraldo-Restelli (Giraldo and Restelli, 2008) | SEM | 50 m | $4^{th}$ | Yes | 14767 |
| NUMA Dyn-SGS (Marras et al., 2015) | SEM | 12.5 m | $4^{th}$ | No | 15056 |
| " | " | 25 m | $4^{th}$ | No | 14992 |
| " | " | 50 m | $4^{th}$ | No | 14535 |
| " | " | 100 m | $4^{th}$ | No | 14325 |
| NUMA SL (Marras et al., 2015) | SEM | 25 m | $4^{th}$ | No | 14918 |
| " | " | 50 m | $4^{th}$ | No | 14726 |
| " | " | 100 m | $4^{th}$ | No | 14551 |
| VMS (Marras et al., 2013) | FEM | 25 m | $1^{st}$ | No | 14890 |
| " | " | 50 m | $1^{st}$ | No | 14629 |
| " | " | 75 m | $1^{st}$ | No | 14487 |
| " | " | 100 m | $1^{st}$ | No | 14355 |
| $f$-wave (Ahmad and Lindeman, 2007) | FV | 50 m | $2^{nd}$ | Yes | 14975 |
| PPM (Straka et al., 1993) | FD | 50 m | | Yes | 15027 |



## Appendix B: Statistics

Since the flow is compressible, we use density-weighed Favre averages following Canuto (1997) when computing horizontally-
averaged statistics. For a scalar $\phi$, the density-weighed average $\bar{\phi}$ at a given height-level $z$ is defined by

$$\bar{\phi} = \frac{\langle \rho\phi \rangle}{\langle \rho \rangle}, \tag{B1}$$

where $\langle \cdot \rangle$ denotes a horizontal mean. All calculations of horizontal statistics are done on the DG nodal mesh to avoid introduc-
ing interpolation errors (Yamaguchi, 2012), with metric terms accounted for in the descriptions of diagnostic variables. The
density-weighted vertical eddy flux for a variable $\phi$ is defined by

$$\overline{w'\phi'} = \frac{\langle \rho\, w'\phi' \rangle}{\langle \rho \rangle}, \tag{B2}$$

where

$$\phi' = \phi - \bar{\phi} \tag{B3}$$

denotes the deviation from the density-weighted mean. The variance can be defined analogously as

$$\overline{\phi'^2} = \frac{\langle \rho\, \phi'^2 \rangle}{\langle \rho \rangle}. \tag{B4}$$

## Appendix C: Hardware

This section summarises the hardware characteristics for the primary compute resources used in tests throughout this paper.
This is particularly relevant to the data presented in Section 7. Compute nodes for the CPU tests were 14-core Intel Xeon
(2.4 GHz), with a maximum memory capacity of 1.5 TB. GPU nodes on this cluster were of 14-core Intel Broadwell (2.4 GHz)
type with 28 cores per node and 256GB memory per node. Compute nodes on the Google Cloud Platform leverage Tesla V100
GPUs available for general-purpose use.

*Author contributions.* **Akshay Sridhar**: analysis; methodology; software; writing–review and editing. **Yassine Tissaoui**: analysis; visualiza-
tion; software; writing–review and editing. **Simone Marras**: conceptualization; methodology; software; writing–original draft preparation,
review and editing. **Zhaoyi Shen**: software; analysis; visualization; writing–review and editing. **Charlie Kawczynski**: software. **Simon
Byrne**: software. **Kiran Pamnany**: software; analysis. **Maciej Waruszewski**: methodology; software. **Thomas H. Gibson**: methodology;
software; writing–review and editing. **Jeremy E. Kozdon**: conceptualization; methodology; software. **Valentin Churavy**: software. **Lucas
C. Wilcox**: conceptualization; methodology; software. **Francis X. Giraldo**: conceptualization; methodology; software; writing–review and
editing. **Tapio Schneider**: conceptualization; methodology; software; project administration; writing–original draft preparation, review and
editing.



*Competing interests.* Simone Marras is a member of the editorial board of Geoscientific Model Development. The peer-review process was
695 guided by an independent editor. The authors have no other competing interests to declare.

*Acknowledgements.* This research was made possible by the generosity of Eric and Wendy Schmidt by recommendation of the Schmidt Futures program, by the Paul G. Allen Family Foundation, Charles Trimble, Audi Environmental Foundation, and the National Science Foundation (grants AGS-1835860 and AGS-1835881). Additionally, V.C. was supported by the Defense Advanced Research Projects Agency (DARPA, agreement HR0011-20-9-0016) and by NSF (grant OAC-1835443). The computations presented here were conducted on the
700 Resnick High Performance Computing Center, a facility supported by Resnick Sustainability Institute at the California Institute of Technology (formerly known as the Central HPC Cluster, with partial support by a grant from the Gordon and Betty Moore Foundation), and on the Google Cloud Platform, with in-kind support by Google. We thank the Google team for their assistance with operations on the Google Cloud Platform. Part of this research was carried out at the Jet Propulsion Laboratory, California Institute of Technology, under a contract with the National Aeronautics and Space Administration.





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
