# Peer review of "Large-eddy simulations with ClimateMachine v0.2.0: a new open-source code for atmospheric simulations on GPUs and CPUs"

_Geoscientific Model Development, 2021_

## Author Comment (AC1)

**GMD 2021-335: Responses to Reviewers**

**March 2022**

We thank the topical editor and both anonymous reviewers for their time and constructive feedback in helping us improve this manuscript. For each statement, RC denotes a reviewer comment, and AC denotes the corresponding author response. Reviewer comments are in black, with author responses in blue. We include specific line, section or page numbers for modified content in the author responses (corresponding to identifiers in the supplementary tracked changes document), where applicable, at the end of each response statement.

**1 Responses to Reviewer 1 (RC1)**

RC: The manuscript describes a new numerical model for simulation of atmospheric flows. The focus is on limited-area high-resolution configurations in large-eddy-simulation-type modeling. The manuscript introduces the new model's background, philosophy, and motivation, describes the model formulation – both the continuous PDEs and discrete system, discusses some representative results, and documents parallel computing performance. Overall, the manuscript is well written, and the presentation is clear. The text is concise and provides a fair amount of detail regarding the model formulation and approximations without compromising accuracy and completeness. The main weakness of the manuscript is the presentation of the results which lack quantitative comparisons. A rigorous and quantitative test case is not presented. Overall, I believe the manuscript is suitable for publication in Geoscientific Model Development. AC: We thank the reviewer for their comments on the scope and suitability of our paper for this journal.

1. RC: The Numerical Experiments (Section 5) provide a nice overview of the model capabilities. Unfortunately, most of the results are presented as contour plots or as comparisons with respect to different model parameter choices (e.g., SGS closure) in the present model. This aspect is the main weakness of the manuscript, and it can be improved. For instance: The Taylor-Green vortex is an exact solution of the Navier-Stokes. For laminar flow the convergence rate can be determined. There are other solutions and/or methods that can be used to quantify the convergence rate of solution error, rather than presenting and comparing contour plots. AC: We agree with the reviewer's comments. We have included measures

for resolution dependence and spatial convergence using the DG discretization described in this paper in section 5.1, lines 342 - 372, and Figure 1. This test problem, the advection of an isentropic vortex, is a part of the ClimateMachine software continuous-integration test suite. We have included a reference to the source code for this case in the data availability section (lines 346-376, Figure 1, and line 633).

2. RC: Some of the numerical experiments are performed as an LES (the turbulence model is very active) and some appear to be (almost) a DNS (no turbulence model in 5.2?). This is not clear. The Density Current appears to be a non-LES calculation. However, it appears from Figure 3 that the flow is not fully resolved because some of the solutions have the tendency to generate small scales. It seems that the numerical method is artificially stabilizing the solution by providing numerical dissipation, resulting in an implicit LES. Can the nature of the simulation be clarified? This has implications relating to the reproducibility of the benchmark. AC: All density current solutions demonstrated in the manuscript have been generated with an active subgrid-scale (SGS) model. In the present version, no DNS results have been presented.

3. RC: The results of Section 6 are somewhat misleading. The results verify the conservation property of prognostic variables. The flux formulation of the numerical method guarantees no internal "leaking" of prognostic variables. However, other types of "energy" are not conserved because the method is dissipative. Perhaps the section should be renamed as "Verification of prognostic variable conservation". This is also stated in the abstract (line 5: "energy-conserving"). "Energy conserving" in numerical methods means that the model conserves second-order moments of prognostic variables, such as kinetic energy and scalar variance and not just the prognostic variables.
AC: We have modified the abstract and section title following the reviewer's suggestion. Both now refer to the conservation of the prognostic variables for clarity. (lines 5, 575)

4. RC: The scaling results are somewhat underwhelming. A maximum of 32 ranks and 16 GPUs is used. A reader might expect that more ranks or GPUs are required by the "ClimateMachine" to simulate Earth's climate. AC: The GPU scaling tests have currently been restricted to the resources available at the time this preprint was produced. They serve as a demonstration of the scaling functionality of ClimateMachine, which is subject to further optimisation in the future.

5. RC: Line 68: Typically, in LES the flow variables are defined as filtered or averaged variables over the volume of the grid cell. A density weighted

average is expected (similar to Appendix B). This should be corrected or clarified.
AC: We have modified the text to clarify that variables in the LES represent resolved large scales. No explicit filtering kernel is applied to the prognostic variables; the resolution of the prognostic variables on the simulation grid with equivalent resolution $\Delta$ implies a filter operation. (lines 273-276)

6. RC: Line 310: Is theta-v the actual buoyancy variable used in the buoyancy gradient in Ri and it is consistent with the energy equation (5)?
AC: Yes, $\theta_v$ is the thermodynamic variable used in this application of the Richardson correction, and is a consistent measure of the effect of buoyancy on the diffusion anisotropy. All thermodynamics functions are included within the code repository; this description of the virtual potential temperature is consistent with the moist phase partitioning computed using the saturation adjustment procedure. (lines 331-333)

7. RC: Typically in LES, the characteristic length scale is modified near the surface, e.g., Mason  Callen (1986, J. Fluid Mech.) is this method applied in the model?
AC: No near-surface modification to the characteristic length-scale is applied in this model. We have added a statement clarifying this to the text in Section 4.2. (line 322-324)

8. RC: Line 285: The deviatoric rate of strain tensor $S_{ij}-1/3\delta_{ij}\text{trace}(S_{ij})$ should be use in the Smagorinsky model, not the rate of strain which has non-zero trace for non-constant density flows.
AC: We have modified the text to clarify the usage of this model in the context of weakly compressible flows typically found in atmospheric LES. We refer to Jähn et al. 2015 and Shi et al. 2018 for recent applications of such a model in the context of LES of atmospheric flows. (lines 306-307)

9. RC: Appendix A3: is this how the BCs are actually applied in the DG method?
AC: Yes, in every application of the boundary conditions, variables at the interior (-) and exterior (+) of the element interface, and the flux vector are exposed to the user through appropriate function arguments. These can then be used to apply boundary conditions in the manner specified in Appendix A3. Further clarity in the code implementation is provided through the code API documentation. Thermodynamic relations are included in the code for appropriate variable conversions.

10. RC: Line 267: "flux tensor" should be "turbulent stress tensor"
    AC: We have modified the phrasing to refer to the "turbulent stress tensor" (line 280).

11. RC: Line 160: is "divergence form" a better term in place of "compact notation"?
    AC: We have modified the phrasing following the reviewer's suggestion (line 171).

12. RC: Line 2: The "performance portability" of the model is not demonstrated in the manuscript.
    We have modified the phrasing to imply scalability, and to clarify the ease of use across CPU/GPU hardware. We have also updated the plain language summary to reflect this change. We believe the results demonstrated in sections 7 and 8 are then consistent with this statement (lines 1-2, 16-18).

13. RC: Line 21: There is a misrepresentation of Smagorinsky (1963) and Lilly (1962) – both here and in other places in the manuscript. These papers do not discuss LES. Smagorinsky (1963) is a pioneering paper about a GCM. Smagorinksy recognized that some form of horizontal dissipation is required to stabilize the GCM since the forward turbulence cascade tends to create smaller scales (similar to Figure 3). He used a simple eddy viscosity parameterization based on the local horizontal rate of strain. Lilly (1962) introduced the TKE parameterization correction for stratified flows which is equation (39) of the current manuscript. The first paper that starts to resemble modern LES is:
    Lilly 1966: On the Application of the Eddy Viscosity Concept in the Inertial Sub-Range of Turbulence, NCAR manuscript 123
    Deardorff in the 70's published several seminal LES papers starting from 1970. A numerical study of three-dimensional turbulent channel flow at large Reynolds numbers. J. Fluid Mech. 41: 453–480. Another useful reference is:
    Smagorinsky, 1993: Some historical remarks on the use of nonlinear viscosities.
    AC: We have modified the text to clarify and highlight the differences in contributions to the dissipation mechanisms in GCMs, and the application of eddy-viscosity models to the interial sub-range of turbulence in modern LES. This is reflected in lines 30-32 (attribution of work), 296 (chronology). While the Smagorinsky (1963) paper refers to a GCM configuration, we believe this is still a pioneering study on subgrid-scale energy transport, detailing techniques which are commonly applied in modern LES (e.g. Germano et al. 1991). (lines 30-32, 299)

**2   Responses to Reviewer 2 (RC2)**

RC: The manuscript covers an interim (as suggested by the version number) report on the development of the atmospheric component of the Earth System modelling suite developed by the CliMA team. The paper reads well. If the goal of the manuscript is to guide potential users (and developers) of the system through the LES functionality of ClimateMachine using a set of examples depicting capabilities of the framework, including achievable scaling, this goal is achieved. Moreover, such a goal is certainly in line with the journal scope. We appreciate the reviewer's comments on the article's broader scope and suitability for this journal.

Such a goal is unfortunately not fully reflected in the "content balance" in the paper. The first ten pages of the work read as a general description of a GCM dynamical core design. While it is of course relevant to subsequent material, numerous introduced aspects of the model are not supported with the examples covered in the paper, e.g., thermodynamics of the ice phase, or "physics" (e.g., precipitation) source terms. Presented examples are discussed more briefly than the not-exemplified model formulation aspects leading to a bit puzzling set of material. Reading through the paper, a question arises: will future papers on ClimateMachine repeat the material from the first ten pages or will they refer to "Large-eddy simulations with ..." for a description of the GCM dynamical core, thermodynamic state description, etc - both options seem undesired.
AC: It is likely that future papers based on the ClimateMachine framework will include additional parameterizations or model terms. In this case, we believe it is best for future articles using this LES framework to address the specific contributions based on their intended scope (whether that is through citation, or a reproduction and expansion of the terms presented in the first 10 pages of this paper).

Why not set up a special issue in GMD (or a GCM/ACP/WCD/ESD/... interjournal SI) devoted to CliMA developments, and extract some of the "commons" from the present work into shorter papers? For instance: (i) the DG numerics with examples supporting their choice; (ii) the thermodynamic variables and examples supporting their choice; (iii) the engineering aspects including the choice of Julia, the parallelisation strategies and benchmark results supporting the choices? Just an idea. Even if the Authors and Editor deem the current content balance OK, perhaps it is worth considering such an option for future publications?
AC: Our intention with the current manuscript is to introduce and present, as a standalone article, the formulation and results from the Discontinuous-Galerkin ClimateMachine Julia large-eddy simulation package. While we appreciate the suggestion to introduce this material as part of a special issue series, we do not believe this is necessary given the limited scope of the present work.

1. RC: The DG numerics are presented as somewhat flawless and trouble-free. Yet, generally, the presented examples do not depict cases of transport of quantities particularly "allergic" to oscillations, smoothing or spurious (negative) values. It would be adequate to extend section 4.3 and at least acknowledge what to expect when using the LES for setups involving chemical and microphysical fields and refer to works discussing it (e.g., Light Durran 2016, doi:10.1175/MWR-D-16-0220.1).
AC: We acknowledge that the DG discretization may present the issues indicated by the reviewer. Section 4.3 now includes references in line 335, to existing numerical stability techniques in DG methods. (Section 4.3, lines 339-342)

2. RC: title (and elsewhere): is the project named ClimateMachine or ClimateMachine.jl?
AC: The project described in this paper is titled ClimateMachine. The ".jl" suffix identifies the project as a Julia package within the code registry, and appears in all references to the code URLs in the manuscript.

3. RC: page 1: "The use of Julia aims to increase accessibility..." - I doubt that employment of a new language with a still minuscule user base https://insights.stackoverflow.com/survey/2021 helps to increase the accessibility. On the other hand, embracing Julia makes adherence to best practices feasible and manageable. As a result, there is a prospect for nurturing modularity, testability and clarity of the code (and the same for its legacy-free dependencies). There are novel coding, debugging, profiling, testing and documentation-generation tools available; the community is vibrant. All this works for the improvement of the code quality and development agility, which will certainly bring benefits to the developers' team and the software users... suggest devoting a separate paragraph to explain to the "average FORTRAN coder" the rationale and expected benefits from shifting to an entirely new simulation-engineering ecosystem, which is a bold and important step.
AC: We have include modifications, including references to recent studies on Julia's feasibility as an HPC language to address the reviewer's comments, with brief statements on the advantages and shortcomings of Julia. We believe this change addresses the reviewer's concern about Julia's relatively small user-base, while highlighting the benefits to physicists of ClimateMachine as a long term research tool, and of Julia as a scientific computing language. (line 16-25)

4. RC: Lilly (1962) and Smagorinsky (1963)? (chronology of paper dates)
AC: We have re-arranged the references in the text to reflect the chronology (line 299).

5. RC: symbol conflict between omega in eq. (4) and domain-defining omega (the bold is merely noticeable)
AC: We have modified the "omega" symbol corresponding to the Coriolis term (now italicized) to distinguish it form the domain-defining "omega". (line 95, 100)

6. RC: Table 1. providing values of constants up to 5 significant digits and providing constants for ice thermodynamics seems unneeded given the scope of the paper. In turn, mentioning the CLIMAParameters.jl package, and the "Overriding defaults" section in its documentation seems more adequate!
AC: We have retained the parameter values in Table 1 for completeness. We have included an additional statement linking Table 1 to the CLIMA-Parameters package in the Code Availability section. We have reduced the number of reported significant digits in Table 1. The full parameter sets are available with the dependent package CLIMAParameters.jl upon installing the ClimateMachine code. (page 6, Table 1)

7. RC: "elements which share boundaries across MPI ranks": this is the very first mention in the text of MPI, ranks, shared boundaries - please first prepare the reader explaining the rank vs. core/CPU/node settings. Mentioning earlier on that the code uses MPI would also make it read better, even if this is "obvious".
AC: We have modified the introduction to highlight the use of MPI in ClimateMachine for clarification (line 17).

8. RC: unclear if the paragraph starting on line 307 on page 12 applies to section 4.2 only or to 4.1 as well
We have included a statement clarifying that this Richardson correction applies to both models described in sections 4.1 and 4.2 (line 333).

9. RC: line 326: it seems more adequate to mention that the Siebesma et al. 2003 case is a non-precipitating boundary-layer shallow-cumulus convection case than bringing up the 1970-ties Barbados experiment origins of the initial thermodynamic profiles used.
AC: We have included a reference to Siebesma et al (2003) following the reviewer's suggestion. We have retained the reference to the 1970 experiment for completeness. (line 550)

10. RC: line 505: mention of Coriolis forcing is likely misleading as it is not the general formulation as given in eq. (4), right?
AC: This is not the general formulation represented in eq(4). We have modified the phrasing to reflect the large-scale pressure gradient effects.

The Coriolis parameter in the large-scale pressure gradient term is consistent with that in Siebesma et al. (2003) (line 553).

11. RC: Figs 7 and 8 are presented in low quality (look like screenshots)
We have reproduced the images at higher resolution. (pages 25,26)

12. RC: line 518: the different behaviour during the first hour spin up (ClimateMachine vs. PyCLES) calls for elaboration.
AC: PyCLES is an anelastic framework with entropy as the prognostic thermodynamic variable. The differences during spinup are attributed to these substantial differences in their model formulation. (lines 566-569)

13. RC: Code availability: please state the licensing terms of the code; line 590: rephrase around "Runtime" (not to confuse with compile-time/runtime)
AC: The code is released under the Apache License, Version 2.0. We have included a statement clarifying this in the Code Availability section. We have also rephrased the installation instructions following the reviewer's suggestion - "Instructions for installing and launching simulations..." (lines 628, 631)

14. RC: NUMA, VMS, PPM acronyms appear only in the table and its caption (NUMA is not even mentioned in the caption), appendix B contains just a single sentence, suggest making the table and its discussion a proper part of the text (and if not, renaming the table from A1 to B1 is likely needed as it is in appendix B, not A).
AC: We have expanded the acronyms in the table caption, and corrected the table identifier. (lines 740, Table A1)

15. RC: there are two appendices labelled B
AC: We have fixed the erroneous title and rearranged the appendices. (lines 721, 739, pages 35-37)

16. RC: some entries have DOIs given, some not; some journal names as abbreviated, some not; title capitalisation is inconsistent (check proper names: taylor-green vortex).
AC: We have updated the references to include DOI links and consistency across title listings. (pages 39-44)

**3 Additional changes**

AC: We have updated the acknowledgements section to correctly reflect contributions from the Heising-Simons Foundation (line 753).

**References**

Germano, Massimo et al. (1991). "A dynamic subgrid-scale eddy viscosity model". In: *Physics of Fluids A: Fluid Dynamics* 3.7, pp. 1760–1765. DOI: 10.1063/1.857955. eprint: https://doi.org/10.1063/1.857955. URL: https://doi.org/10.1063/1.857955.

Jähn, M. et al. (2015). "ASAM v2.7: a compressible atmospheric model with a Cartesian cut cell approach". In: *Geoscientific Model Development* 8.2, pp. 317–340. DOI: 10.5194/gmd-8-317-2015. URL: https://gmd.copernicus.org/articles/8/317/2015/.

Shi, Xiaoming et al. (2018). "An Evaluation of LES Turbulence Models for Scalar Mixing in the Stratocumulus-Capped Boundary Layer". In: *Journal of the Atmospheric Sciences* 75.5, pp. 1499 –1507. DOI: 10.1175/JAS-D-17-0392.1. URL: https://journals.ametsoc.org/view/journals/atsc/75/5/jas-d-17-0392.1.xml.